# Robustness of a parsimonious subsurface drainage model at the French national scale

Alexis Jeantet[1], Hocine Hénine[1], Cédric Chaumont[1], Lila Collet[1,2], Guillaume Thirel[1], Julien Tournebize[1]

[1]INRAE, University of Paris Saclay, UR HYCAR, 1 rue Pierre Gilles de Gennes, Antony, France

[2]Now at: EDF R&D, OSIRIS Department, 7 boulevard Gaspard Monge, 91120 Palaiseau, France

*Correspondence to*: Alexis Jeantet (alexis.jeantet@inrae.fr)

**Abstract -** Drainage systems are currently implemented on agricultural plots subjected to temporary or permanent waterlogging issues. Drained plots account for 9% of all arable soils in France. As such, the need for accurate hydrological modeling is crucial, especially in an unstable future context affected by climate change. The aim of this paper is to assess the capacity of the SIDRA-RU hydrological drainage model to represent the variability of pedoclimatic conditions within French metropolitan areas, as well as to demonstrate the utility of this model as a long-term management tool. The model is initially calibrated using the KGE' criterion as an Objective Function (OF) on a large and unique database encompassing 22 plots spread across France and classified according to three main soil textures (silty, silty-clayey and clayey). The performance of SIDRA-RU is evaluated by monitoring both the set of KGE' calibration values and the quality of simulations on each plot with respect to high and low discharges as well as the annual drained water balance. Next, the temporal robustness of the model is assessed by conducting on selected plots the split-sample test capable of satisfying the data requirements. Results show that the SIDRA-RU model accurately simulates drainage discharge, especially on silty soils. The performance on clayey soils is slightly weaker than that on silty soils yet remains acceptable. Similarly, the split-sample test indicates that SIDRA-RU is temporally robust on all three soil textures. Consequently, the SIDRA-RU model closely replicates the diversity of French drained soil and could be used for its long-term management potential.

## 1. Introduction

Subsurface drainage is an agricultural soil management technique that controls soil water content and increases aeration on soils subjected to temporary or permanent water saturation issues into the soil depth (Jamagne, 1968; Baize and Jabiol, 2011). Plot water conditions are stabilized, thus ensuring better crop yields (Broadhead and Skaggs, 1982; Armstrong et al., 1988; Nijland et al., 2005; Ibrahim et al., 2013) while reducing the flood risk on plots (Henine et al., 2014; Tuohy et al., 2018b). Drained soils often belong to the hydromorphic soil category and sometimes in French context lie on a shallow and impervious layer reducing the deep infiltration (Thompson et al., 1997; Lange et al., 2011).

In France, all artificially drained soils comprise more than 2.7 million hectares of arable soils (source: "RGA - Agreste" (2010)), i.e. close to 10% of all arable land, corresponding to about 20% for cereal-type field crops. In practice, several techniques exist to drain soils such as subsurface drainage and open ditch. However, in France, over 80% of drainage practices are conducted by introducing perforated pipes lying on the impermeable layer. The drain depth, spacing, slope and diameter of these pipes constitute the main characteristics of each design; they are constrained by the local study site conditions, such as soil characteristics and climate (Mulqueen, 1998).

Since the economic and environmental consequences of climate change are of increasing concern to stakeholders, proper drainage practices have become a major issue. Predicting the long-term behavior of these systems is even more crucial in this context of water resource protection and restoration since drainage has an impact on water quality (Tournebize et al., 2012, 2017, 2020). The literature contains studies targeting the impact of climate change on drainage practices, with an
emphasis on either the increase in annual drained water balance (Pease et al., 2017) or agricultural productivity on drained plots (Jiang et al., 2020a). These topics raise concerns over the sustainability of existing drainage systems, and their need to be redesigned has come to the fore (Deelstra, 2015; Abd-Elaty et al., 2019). A common theme across all these studies is the need to properly represent drainage systems within each study area.

In this context, hydrological modeling offers a widespread tool for predicting drainage discharge, with several models
currently in use, e.g. DRAINMOD (Skaggs, 1981; Skaggs et al., 2012) in the United States. This spatially-distributed model operates on various spatial scales (Konyha and Skaggs, 1992; Brown et al., 2013) and integrates many modules in order to represent different hydrological processes and solute transports (Breve et al., 1997). In Europe, the MACRO model (Larsbo et al., 2005; Jarvis and Larsbo, 2012) is currently used by the FOCUS group (Adriaanse et al., 1996; Boesten et al., 1997) to evaluate drainage system performance and contaminant transport (Jarvis et al., 1997; Beulke et
al., 2001). These two models, despite demonstrating their effectiveness, have been designed using physically-based modeling strategies with large number of parameters. Their calibration on several study sites becomes difficult and time consuming (Beven, 1989).Given this complexity, the SIDRA-RU model offers an interesting alternative. The model is semi-conceptual (Beskow et al., 2011), being composed of one physically-based part (the SIDRA module) coupled with a conceptual part (the RU module) and parsimonious, by virtue of requiring the calibration of only six parameters, hence
making it easy to configure (Perrin et al., 2003). Initially intended to simulate drained discharge during flood periods (Lesaffre and Zimmer, 1987a), the SIDRA model converts weather-dependent soil recharge into drainage discharge by solving a semi-analytical formula derived from the Boussinesq equation (Boussinesq, 1904). Various modules have been integrated so as to better represent infiltration (Kao et al., 1998), water flux in the unsaturated zone (Bouarfa and Zimmer, 2000) or pesticide leaching (Branger et al., 2009). The RU module was recently integrated in order to model water transfer
in the unsaturated zone (Henine et al., in review). This new version inputs a continuous recharge term into the SIDRA module, which then allows for the simulation of drainage discharge over the entire hydrological cycle.

Due to soil diversity within French drained areas, a model used for management purposes must initially be as general as possible and correctly calibrated to ensure model behavior matches the behavior of each studied site as closely as possible (Perrin, 2000). As such, a reliable calibration protocol often depends on the choice of Objective Function (OF), which
serves as the numerical criterion to be optimized so that the simulation more accurately resembles reality. Many OFs can be used to calibrate a model, such as the Nash-Sutcliff Efficiency (NSE, Nash and Sutcliffe (1970)) or the Kling-Gupta Efficiency (KGE, Gupta et al. (2009)), depending on the purpose of the particular study. The model here is intended for use on future prediction data under a long-term management scenario. From this perspective, the model must be temporally robust, i.e. its performance and parameters must remain independent of the period chosen for calibration
(Klemeš, 1986). Such an evaluation can be performed by means of various tests, which tend to depend on the model structure (Refsgaard and Storm, 1996; Refsgaard, 2001; Henriksen et al., 2003; Daggupati et al., 2015). Since SIDRA-RU is a simple model, the split-sample test (Klemeš, 1986) is considered to be sufficient (Refsgaard, 1997). However, the national-scale evaluation of a hydrological model requires a large database, which is not readily available in the drainage hydrology field. This lack of data is the reason for the paucity of studies in the current literature.

The aim of this study is to assess the ability of a hydrological drainage model to simulate observed drainage discharge across several representative sites spread out in France. An exhaustive database, composed of 22 experimental sites and encompassing the main drained regions in France, has been built to account for the large diversity of French drained soils on which the model was tested. Database completeness is one of this paper's main strengths and allows generalizing our results on soil diversity. The hydrological model chosen for this work is SIDRA-RU, a parsimonious model that yields continuous simulations and can easily be run on the database. In addition, the SIDRA-RU model offers a novel tool for the hydrological drainage modeling community; this study therefore provides an opportunity to test its performance at the national scale, which raises another point of interest regarding this study. Moreover, the temporal robustness of the SIDRA-RU model is assessed in the aim of asserting whether or not the model can be used within the scope of a long-term management tool, i.e. one capable of incorporating climate change.

## 2. Materials and methods

### 2.1 French classification of drained soils

A multitude of materials constitute French soils, as defined by their geological context, textural evolution and regional climate. All of the above characteristics serve to determine the uniqueness of a soil. Making generalizations about soil diversity becomes then a necessary step. Indeed, grouping them by soil category facilitates their modeling. Several official classifications serve to group soil types (FAO, 1988; Krogh and Greve, 1999; Driessen et al., 2000). In this study, we are proposing to classify them by texture, thus making it possible to sort the database into three categories (see Fig. 1 & Table 1). Let's note that here we do not consider the geological context or the regional climate to classify soils.

The Lagacherie and Jamagne classification (Jamagne et al., 1977; Lagacherie and Favrot, 1987; Richer-de-Forges et al., 2008) has been used to evaluate this strategy. According to Fig. 1 (top), three distinct soil types occupy most of the regions with the highest drainage ratio (i.e. percentage of a land area that has been drained, with drainage ratio values above 50% of total arable area). First, the glossic and planolosic soils, belonging to the Luvisols and mainly located around the Paris Basin and in the Allier Region (see Fig. 1), are defined by textural differentiation between the surface horizon, which is often silty and sometimes sandy-loamy, and a deep clayey horizon. Second, brown acidic and leached soils, mostly distributed in the western part of France (Fig. 1), lie on a magmatic and metamorphic substratum; they are often characterized by a silty-clayey or loamy texture. Third, the pseudogley soils are also substantially drained, yet they remain only slightly correlated with any specific soil texture. We assume here that among the 3 studied soil textures, they are more related to silty-clayey soils (Lévy, 1972). Figure 1 (bottom) shows the drainage ratio of arable soil in France. Most drainage systems (approximately 80%) lie on a loamy texture, according to Lagacherie and Favrot (1987), except in the eastern part of France (Fig. 1), where drained soils are either predominantly silty-clayey or heavy clayey soils.

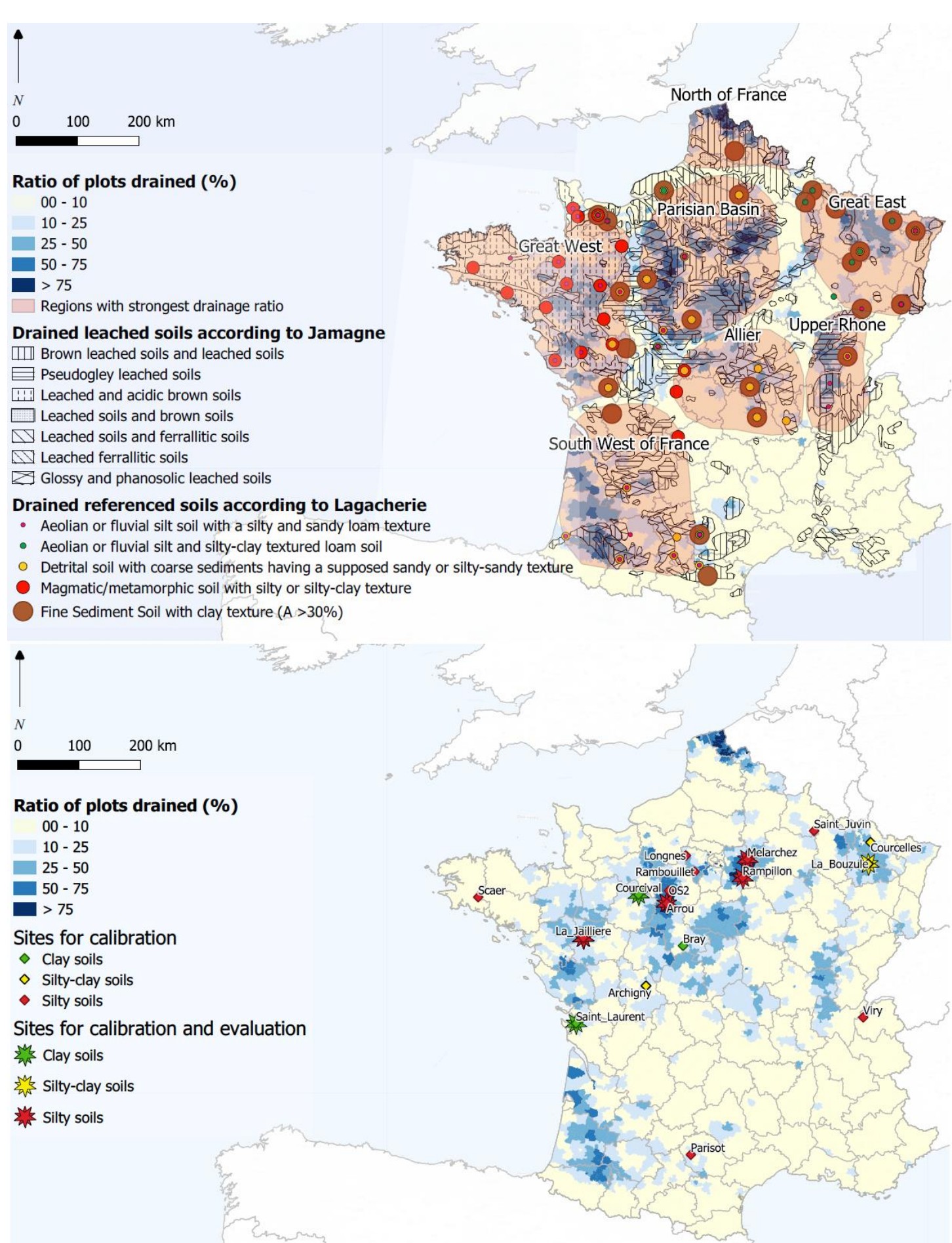

*Figure 1: top) the pedological distribution of French soils, produced by Jamagne et al. (1977) and Lagacherie and Favrot (1989), overlaid on the spatial distribution of French drainage (RGA, 2010); bottom) the spatial distribution of sites aggregated by texture with observed flow data*

108
109

*Table 1: Main characteristics of the 22 plots with observed discharges and associated KGE' from the calibration process and split-sample tests*

| Texture | Plot name | Site location | Data period (years) | Surface area (ha) | Tile depth (m) | Space between drain and mid-drain[1] (m) | Use | KGE' from total calibration | KGE' from the Split-Sample Test In calibration P1 | In calibration P2 | In evaluation P1 | In evaluation P2 | Index (Fig. 10) |
|---|---|---|---|---|---|---|---|---|---|---|---|---|---|
| Clayey soil | Bray_P1 | Bray | 6 | 46.0 | 0.9 | 5.0 | calibration only | 0.52 | - | - | - | - | - |
| | Courcival_P3 | Courcival | 10 | 1.3 | 0.9 | 6.0 | calibration and evaluation | 0.44 | 0.42 | 0.48 | 0.36 | 0.34 | C1 |
| | La_Bouzule_P1 | La Bouzule | 13 | 1.9 | 0.9 | 12.0 | calibration and evaluation | 0.52 | 0.58 | 0.52 | 0.42 | 0.31 | C2 |
| | Saint_Laurent_P2 | Saint Laurent | 15 | 1.2 | 0.9 | 10.0 | calibration and evaluation | 0.76 | 0.75 | 0.75 | 0.58 | 0.63 | C3 |
| Silty soil | Arrou_P6 | Arrou | 19 | 2.0 | 0.8 | 5.0 | calibration and evaluation | 0.65 | 0.57 | 0.78 | 0.75 | 0.55 | S1 |
| | Arrou_P8 | Arrou | 19 | 2.0 | 0.8 | 10.0 | calibration and evaluation | 0.65 | 0.54 | 0.72 | 0.72 | 0.53 | S2 |
| | Courcelles_P1 | Courcelles | 3 | 2.4 | 0.9 | 5.0 | calibration only | 0.70 | - | - | - | - | - |
| | Courcelles_P2 | Courcelles | 3 | 2.4 | 0.9 | 5.0 | calibration only | 0.68 | - | - | - | - | - |
| | La_Jaillière_P4 | La Jaillière | 16 | 1.0 | 0.9 | 5.0 | calibration and evaluation | 0.83 | 0.82 | 0.87 | 0.79 | 0.82 | S3 |
| | Longnes_P2 | Longnes | 6 | 1.0 | 0.9 | 6.0 | calibration only | 0.66 | - | - | - | - | - |
| | Longnes_P3 | Longnes | 6 | 1.0 | 0.9 | 7.5 | calibration only | 0.58 | - | - | - | - | - |
| | Melarchez | Melarchez | 39 | 700.0 | 0.9 | 6.0 | calibration and evaluation | 0.77 | 0.77 | 0.75 | 0.61 | 0.61 | S4 |
| | OS2_P1 | OS2 | 3 | 2.0 | 1.0 | 5.0 | calibration only | 0.54 | - | - | - | - | - |
| | Parisot_P2 | Parisot | 4 | 0.3 | 0.9 | 6.0 | calibration only | 0.81 | - | - | - | - | - |
| | Rambouillet_P1 | Rambouillet | 3 | 1.0 | 0.9 | 5.0 | calibration only | 0.77 | - | - | - | - | - |
| | Rampillon_GP | Rampillon | 9 | 355.0 | 0.9 | 6.0 | calibration only | 0.77 | 0.80 | 0.76 | 0.71 | 0.73 | S5 |
| | Saint_Juvin_P1 | Saint Juvin | 3 | 1.4 | 0.9 | 15.0 | calibration only | 0.56 | - | - | - | - | - |
| | Scaer_P2 | Scaer | 5 | 1.4 | 0.9 | 5.0 | calibration only | 0.61 | - | - | - | - | - |
| | Viry_P1 | Viry | 4 | 0.8 | 0.9 | 7.5 | calibration only | 0.60 | - | - | - | - | - |
| Silty-clayey soil | Archigny_P1 | Archigny | 2 | 1.0 | 0.9 | 4.0 | calibration only | 0.76 | - | - | - | - | - |
| | Archigny_P2 | Archigny | 2 | 1.0 | 0.9 | 6.0 | calibration only | 0.76 | - | - | - | - | - |
| | La_Bouzule_P2 | La Bouzule | 13 | 2.9 | 0.9 | 6.0 | calibration and evaluation | 0.54 | 0.50 | 0.60 | 0.45 | 0.56 | SC1 |

---

[1] The midpoint between consecutive drains

## 2.2 Input data

A representative database of drainage discharge across France was specially assembled for this study. The data originate from various sources: (1) the ORACLE research project (Tallec et al., 2015) and artificial Rampillon wetland (Tournebize et al., 2012, 2017; Lebrun et al., 2019); (2) the partnership with the ARVALIS Institute, which monitors the *La Jaillière* experimental site; and (3) data from reference drainage sites dating between the 1960's and 1980's (Lagacherie and Favrot, 1987; Jannot, 1988).

The data from this last source stem from monitoring experiments managed by INRAE (formerly Cemagref) that test drainage modalities, i.e. what depth, space or pipes best fit to the field conditions. The combination of these sources yields a database of nearly 200 years of cumulative hydrological records on drained plots in diverse pedoclimatic contexts over a broad plot scale range (e.g. 0.8-700 ha). The resulting extensive dataset compiled for hydrological modeling purposes encourages the transferability of this study's findings.

The drainage network of the various study plots is based on similar technical characteristics (see Table 1), composed of PCV perforated and corrugated pipes lying on a depth of 0.85 m to 1 m, with an inter-drain spacing from 8 to 24 m. The most widely used method for monitoring drainage discharge consists of measuring the corresponding water level at the drainage collector outlet using a calibration curve fitted at each measurement site, by designing a control section where flow is hydraulically managed. Before the 1980's and 1990's, data were recorded on a paper sheet that followed the motion of a floater linked to the water level. Nowadays, water level sensors (floating systems equipped with ultrasonic measurements) are used and the data are digitally recorded. To ensure data homogeneity, observed data have been manually assessed by expert judgment in order to highlight periods of suspect data quality; as deemed necessary, the data have been corrected or deleted. The 22 study plots are distributed over three distinct soil textures: silty, silty-clayey, and clayey. Fifteen of them are characterized by a silty soil texture, in covering most French regions (Fig. 1). The database is more limited as regards clayey soils, characterized by sites like *Saint_Laurent_P2* or *Courcival_P3*. Some regions, e.g. eastern France, which are strongly characterized by a clayey texture yet with just one clayey site, are not well represented. The SIDRA-RU performance in this region will be estimated from the SIDRA-RU global performance on clayey soils from the database, comprising 44 years of observed discharges (Table 1). Moreover, some regions with a high drainage ratio, e.g. southwestern France, have no observation points and are therefore not covered by this study. Lastly, the pseudogley soils are mostly correlated with the silty-clayey soils, yet the database does not provide any relevant silty-clayey plots. Model performance will thus be estimated by the global performance for all such sites. Each site was defined by the aforementioned technical characteristics (drain depth, mid drain spacing corresponding to the half-space between two successive drains, surface area), plus the length of available observed discharge logs and suitability to the split-sample test (Table 1).

Due to a lack of agronomic data, we assume here that growing practices do not affect the subsurface drainage hydrology on the study plots, except in the absence of a tillage technique (Dairon et al., 2017), a situation that is not widespread in France. This assumption is supported by four observations, namely: 1) the subsurface drainage is mainly effective during fall and winter, when the actual evapotranspiration is low; 2) our study investigates drained soils in winter primarily used for winter crops such as wheat and maize (Zimmer, 1996), so we assume that the studied plots are cultivated every year throughout this season without a fallow period; 3) these crop types impact the subsurface drainage hydrology in a similar manner; hence, annual crop rotation does not add significant bias to the model calibration; and 4) the effect of cover crops

(Meyer et al., 2018) has been neglected due to the fact that they were not widely used before 2012 and, for 19 of the 22 sites, the corresponding study periods ended before 2012.

The meteorological data were provided by the SAFRAN database (Vidal et al., 2010), a meteorological reanalysis covering France and supplying both precipitation and potential evapotranspiration (PET, based on the FAO-56 Penman-Monteith PET (Córdova et al., 2015)) data on all database-referenced plots. These data are available from 1959 to 2019 at a daily time step and a spatial resolution of 8 km, which is 1,000 times greater than the scale of the studied plots. This difference may introduce errors on model outputs, but all such errors are considered to be negligible.

### 2.3 The SIDRA-RU model

The SIDRA-RU model is a semi-conceptual, lumped model that describes the hydrological processes of artificial drainage systems. This model is based on the principle of rainfall-drainage discharge conversion and uses the rainfall P and potential evapotranspiration PET to predict water table height and drainage discharge at the drainage network outlet.

Three modules have been integrated into the current version of SIDRA-RU (see Fig. 2):

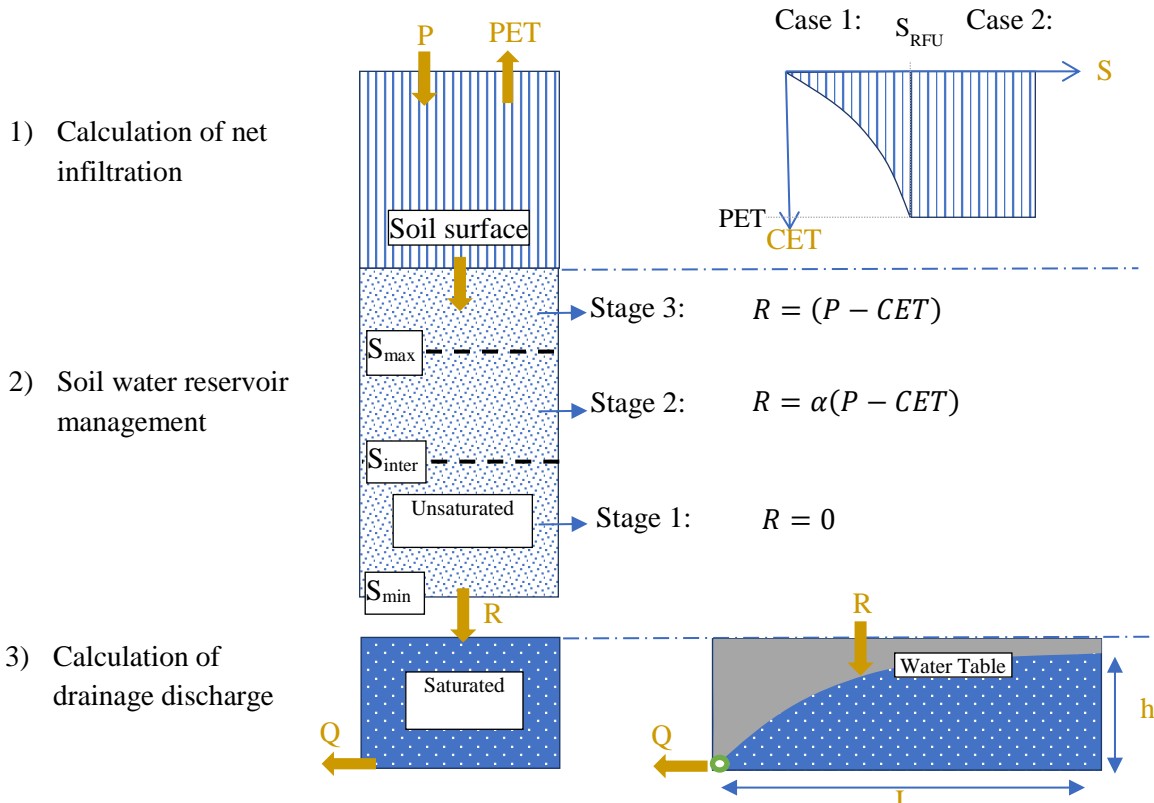

*Figure 2: Diagram presenting the various modeling stages of the SIDRA-RU model*

First, an evapotranspiration module converts PET into an approximate value of actual evapotranspiration, called corrected evapotranspiration (or CET), from the available water level S(t) in storage to satisfy the evapotranspiration constraint. A $S_{RFU}$ threshold is set, thereby assigning the minimum water level to fully satisfy PET (see Eq. (1)):

$$CET(t) = \begin{cases} PET(t) * e^{-\frac{S_{RFU}-S(t)}{S(t)}} & if \ S(t) < S_{RFU} \\ PET(t) & if \ S(t) \geq S_{RFU} \end{cases} \tag{1}$$

The net infiltration $P_{net}(t)$ is calculated by subtracting CET(t) from P(t).

Second, the RU module, a conceptual storage, calculates the water table recharge term R(t) (mm) from the meteorological input and water storage capacity of the soil reservoir. Two main parameters control this RU module. On the one hand, the $S_{inter}$(mm) parameter serves as an intermediate threshold of the soil reservoir defining the water quantity required to generate flow in the reservoir before saturation of the storage (see Eq. (2)):

$$S_{RFU} = a * S_{inter} \tag{2}$$

Here, the parameter $a$ is set to 0.4 due to the water capacity easily available for use by the crops (RFU, for "Réserve Facilement Utilisable" in French), representing approximately 60% of $S_{inter}$ (approximate concept of water holding capacity) on French drained soils (Tournebize et al., 2015). On the other hand, the $S_{max}$ (mm) parameter represents the maximum capacity of the soil reservoir from which the net infiltration is fully converted into R(t). These two parameters constitute an approximate concept of the water holding capacity of a soil. Three stages are to be considered (Fig. 2):

- Stage 1: $S(t) < S_{inter}$, the water level is too low to allow for the generation of subsurface flows to the drains, i.e. Eq. (2):
$$R(t) = 0 \; ; \; S(t) = S(t-1) + P_{net}(t) \tag{3}$$
- Stage 2: $S(t) \in [S_{inter}; S_{max}[$, the water level is high enough to partially allow for the generation of water table recharge R(t). The parameter $\alpha$ defines the proportion of $P_{net}(t)$ being converted to recharge R(t), while the remainder updates the water level, i.e. Eq. (3):
$$R(t) = \alpha * P_{net}(t) \; ; \; S(t+1) = (1-\alpha)\, P_{net}(t) + S(t) \tag{4}$$
A sensitivity analysis on the SIDRA-RU model has revealed that $\alpha$ is not sensitive to the KGE' criterion (Henine et al., in review), used in this study as OF (see section 2.4.1), and moreover can be set at $1/3$. Hence, to limit uncertainties relative to the calibration process for a non-sensitive parameter, this approach has been conserved herein.

Stage 3: $S(t) \geq S_{max}$, water storage is full, i.e. Eq. (4):
$$R(t) = P_{net}(t) = P(t) - CET(t). \tag{5}$$

Third, the calculated water table recharge R(t) feeds the original SIDRA module (Lesaffre and Zimmer, 1987b; Bouarfa and Zimmer, 2000) in order to calculate the water table level h(t) and drainage discharge Q(t) (see Eqs. (5) and (6)), in solving a semi-analytical formula derived from the Boussinesq equation (Boussinesq, 1904). This physically-based module is mainly controlled by two parameters: the horizontal hydraulic conductivity K(m/d), and drainage porosity μ(-).

$$\frac{dh(t)}{dt} = \frac{R(t) - K\frac{h(t)^2}{L^2}}{A_2 \mu} \; ; \; h(t+1) = h(t) + \frac{dh(t)}{dt} \tag{6}$$

$$Q(t) = AK\frac{h(t)^2}{L^2} + (1-A)R(t) \tag{7}$$

- L: mid drain spacing (m);
- $A_2$: second water table shape factor, $A_2 \approx 0.89$ (Lesaffre, 1989);
- A: third water table shape factor, $A = 0.869$ (Lesaffre, 1989). We are supposing here that the water table shape between the drain and mid-drain is an ellipse. A is therefore obtained by integrating ¼ of this reference ellipse (see Fig. 2, Part 3: Calculation of drainage discharge).

It can be noted that surface runoff is considered to be negligible in the model, only contributing slightly to total flow (Kuzmanovski et al., 2015). Furthermore, one of the assumptions made in the SIDRA module was to consider that pipes

lie on an impervious layer, thus all excess water is fully released through the pipes (Lesaffre, 1989). This assumption seems rather reasonable since a large majority of French drained sites lie on such soils, according to the studies carried out on the aforementioned drainage reference areas (Lagacherie and Favrot, 1987; Jannot, 1988; Tournebize et al., 2012; Tallec et al., 2015; Tournebize et al., 2017; Lebrun et al., 2019). To be completely operational, SIDRA-RU requires information on technical characteristics, such as drain depth P(m) and mid drain spacing L(m) between the drain and inter-drain. Furthermore, due to the aforementioned assumptions dealing with the parameters $a$ and $\alpha$, a calibration process is only necessary for four parameters: K and μ, with the $K/\mu$ ratio describing the responsiveness of the system, and $S_{inter}$ and $S_{max}$ from the RU module.

## 2.4 Methods

### 2.4.1 Calibration method

Parameter optimization (i.e. calibration) is commonly performed in a hydrological modeling context in order to adapt the hydrological model parameters to the specific study area context. This process is relevant for conceptual parameters since, by construction, they cannot be measured directly nor easily correlated with any physical characteristics of the studied system. Some of the physically-based parameters that are too difficult to measure can also be calibrated.

The model calibration herein has been based on the algorithm implemented in the "airGR" R package (Coron et al., 2017, 2020), and is composed of two parts. First, a systematic examination of the parameter space provides the most likely zone of convergence, on the basis of a grid-screening algorithm (Mathevet, 2005), according to a given performance criterion (i.e. Objective Function, OF). Each parameter space is defined by its specific distribution and intrinsic statistical characteristics, with respect to the soil texture. Second, a steepest-descent local search procedure (Michel, 1991) seeks to improve the OF, beginning with the grid-screening part, and find a more accurate estimate of the parameter set, i.e. with higher model performance.

The hydraulic conductivity K and drainage porosity μ follow a log-normal distribution (Rousselot and Peyrieux, 1977; Kosugi, 1994, 1996, 1999; Rousseva et al., 2017; Ren and Santamarina, 2018). Parameters $S_{inter}$ and $S_{max}$ are conceptual and thus not defined by an intrinsic distribution. However, they are similar to the water holding capacity of a soil, which follows a normal distribution (Vachaud et al., 1985; Brocca et al., 2007; Biswas et al., 2012; Biswas, 2019); consequently, in this study, both $S_{inter}$ and $S_{max}$ are described as following a normal distribution. The mean and standard deviation of each parameter are available in Appendix A. The ones for K and μ were extracted per soil texture from the aforementioned reference drainage areas. The ones for $S_{inter}$ and $S_{max}$ were numerically fixed after many calibration tests.

Various OFs are commonly used in hydrological calibration processes, depending in large part on the primary aim of the study. The most widespread OFs are RMSE (Anderson and Woessner, 1992), MSE (Ye et al., 2020), NSE (Nash and Sutcliffe, 1970) and, more recently, KGE (Gupta et al., 2009). Our goal here is to evaluate model performance in order to simulate an entire hydrological cycle and represent the inter-annual variations of the study plot to produce long-term projections about future drainage hydrology. We have thus introduced the KGE' criterion (Kling et al., 2012), an evolution of KGE that is more relevant than NSE in reproducing internal flow rate variability (Santos et al., 2018). KGE' is defined by three modeling error components, as combined in Eq. (8):

$$KGE' = 1 - \sqrt{(r-1)^2 + (\beta-1)^2 + (\gamma-1)^2} \qquad (8)$$

with:

- $r = \frac{cov(Q_{obs}, Q_{sim})}{\sigma_{obs}^2 \sigma_{sim}^2}$: the Pearson correlation coefficient, which serves to evaluate the error in both shape and timing between observed and simulated flows, with $cov$ being the covariance between observed and simulated flows and $\sigma$ their standard deviation;

- $\beta = \frac{\mu_{sim}}{\mu_{obs}}$: the bias term, which evaluates the bias between observed and simulated flows, with $\mu$ being the mean of observed and simulated discharges, respectively;

- $\gamma = \frac{\mu_{obs}\sigma_{sim}}{\sigma_{obs}\mu_{sim}}$: the ratio between observed and simulated coefficients of variation, which serves to evaluate the flow variability bias.

KGE' values range from $-\infty$ to 1. The model performance improves as KGE' increases towards 1. If the reader intends

to use the mean flow benchmark as a reference (corresponding to NSE = 0) in order to assess KGE', the target value is KGE' = -0.41 (Knoben et al., 2019). During the model calibration step, the data series over the whole time period was used for each studied plot (Table 1).

### 2.4.2 Split-sample test

The temporal robustness of the model proves to be of utility when facing a time period different from that chosen for calibration, i.e. enhancing the model's capacity to perform equally well over different and contrasted time periods (Li et al., 2011). This point is particularly important when the model is intended for application under future climate change scenarios (Thirel et al., 2015b).

The choice of evaluation strategy mainly depends on model structure. For lumped conceptual models, a simple split-
260 sample test (Klemeš, 1986) is sufficient to assess the robustness of such a model (Refsgaard and Storm, 1996; Daggupati et al., 2015). The split-sample test, as illustrated in Fig. 3, consists of splitting the data period into two sub-periods (P1 and P2) and then calibrating the model over both of them independently.

Thus, two optimal parameter sets are obtained (one covering P1 the other P2), with each being tested over the other sub-period (e.g. evaluation sub-period P2 for a calibration over sub-period P1). If the KGE' scores from the evaluation sub-
265 period lie close to the calibration KGE' score, then the model calibration is considered as temporally robust and independent of the chosen time period.

This test was performed on the records from 9 plots showing at least 10 years of time-series data (Table 1). These series were split into two equal-length periods, and the KGE' scores produced over the calibration and evaluation sub-periods were assessed and compared.

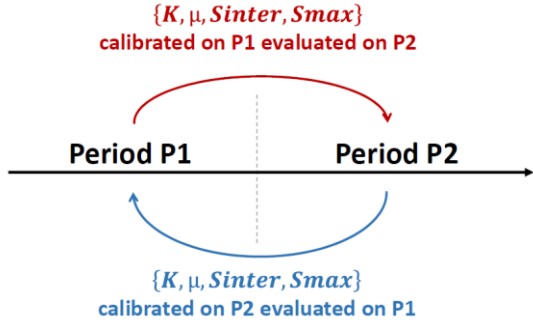

*Figure 3: Functional diagram of the split-sample test*

### 2.4.3 Numerical evaluation criteria

Evaluating and highlighting model performance limitations during the calibration process using numerical scores, such as NSE, RMSE or KGE criteria, is often recommended, in addition to graphical analysis (Moriasi et al., 2015). However, understanding their evolution is indeed difficult. To overcome this difficulty, a model performance classification by means of a range of numerical score values will help assess the calibration quality.

In subsurface drainage modeling, the bibliography is not exhaustive with respect to KGE' score ranges. The few articles published deal instead with catchment hydrology modeling (Crochemore et al., 2015; Poncelet et al., 2017). Use of the NSE criterion has been detailed more extensively (Moriasi et al., 2007; Ritter and Muñoz-Carpena, 2013; Moriasi et al., 2015); moreover, studies focusing on model calibration using the NSE score in subsurface drainage modeling state that values above 0.5 are considered to be acceptable (Helwig et al., 2002; Wang et al., 2006; Tuohy et al., 2018a). Even if a comparison between the NSE and KGE' scores is theoretically incorrect and not unequivocal (Criss and Winston, 2008; Knoben et al., 2019), we are assuming herein that the score ranges using the NSE criterion can be transposed to those using the KGE' criterion. We have decided to qualify the KGE' values as follows: the model calibration using KGE' values greater than or equal to 0.5 leads to acceptable model performance. KGE' values ranging from 0.6 to 0.7 are considered to reflect good performance, while a KGE' greater than or equal to 0.7 is deemed very good performance (Table 2).

The model has also been evaluated in terms of its capacity to reproduce annual cumulative discharges, with a direct comparison conducted between observed and simulated flow rates using both the linear correlation coefficient $R^2$ (Bailly and Carrère, 2015) and the associated linear regression equation.

## 3. Results

### 3.1 Model performance after calibration

Table 1 lists the performance over the entire calibration period obtained from all 22 sites and Table 2 classifies the model performance from each soil texture according to the score ranges. Performance varies across the three soil textures, with both unsatisfactory KGE' values, e.g. for the *Courcival_P3* site, and some "very good" KGE' values, e.g. *Parisot*. For 21 of the 22 referenced plots, the calibration KGE' lies above 0.5, thus revealing at least "acceptable" KGE' values. The silty plots show values ranging from 0.54 to 0.83, including the best model performances, such as *La_Jaillière_P4* plot, with a KGE' of 0.83. They include three "acceptable" scores, reaching "good" for six of them and "very good" for another six. The silty-clayey plots exhibit relatively homogenous KGE' values, ranging from 0.54 to 0.76. As regards the clayey plots, KGE' values display a wider range than on the silty-clayey plots, i.e. from 0.44 at the *Courcival_P3* plot to 0.76 at *Saint_Laurent_P2* but the model performance remains at least "acceptable" on most of them (including one "very good"). *Courcival_P3* is the only one indicating an "unsatisfactory" KGE' value.

*Table 2: KGE' calibration scores*

| KGE' value range | Scores | Number of plots | Silty soils | Silty-clayey soils | Clayey soils |
|---|---|---|---|---|---|
| < 0.50 | Unsatisfactory | 1 | - | - | 1 |
| [0.50 - 0.60[ | Acceptable | 6 | 3 | 1 | 2 |
| [0.60 - 0.70[ | Good | 6 | 6 | - | - |
| ≥ 0.70 | Very good | 9 | 6 | 2 | 1 |
| Total | - | 22 | 15 | 3 | 4 |

The *La_Jaillière_P4* plot is used as an example to illustrate the temporal comparison between observed and simulated discharges over 16 years (see Fig. 4). These same graphs are available in Appendix B to illustrate the case of a silty-clayey soil at *La Bouzule_P2* and in Appendix C for the case of a clayey soil at *Saint-Laurent_P2*.

Figure 4 shows that the simulated discharges are in good agreement with observations in terms of both seasonal dynamics (dry and wet season alternation) and cumulative distribution. Rainfall series are not directly correlated with discharge, as some rainfall events appearing from September to November do not systematically lead to subsurface flow. However, winter rains typically turn into discharge after a period of one or two days. A graphical analysis shows that simulated drainage discharges generally start in the same period as the observed discharge, with various delays depending on soil type.

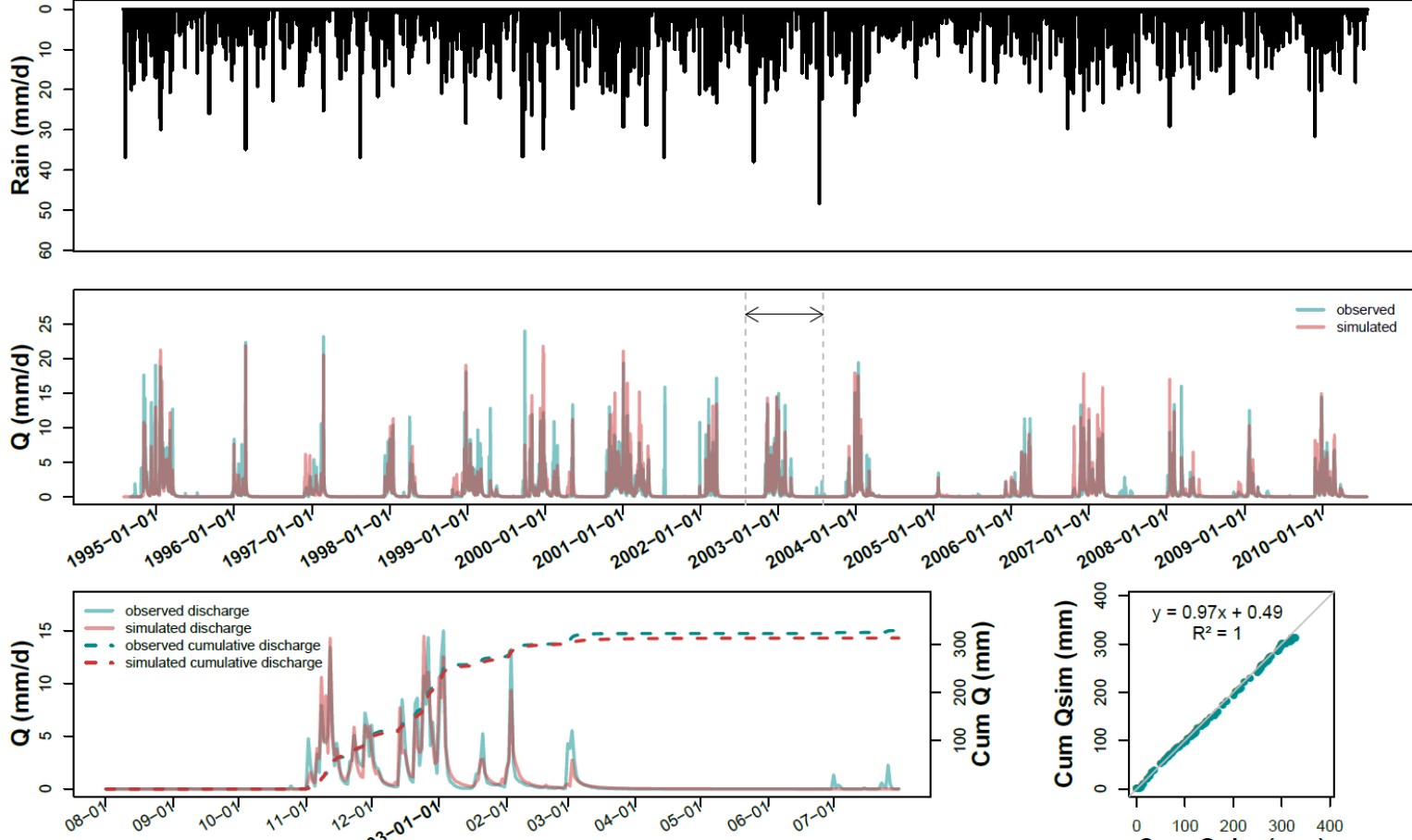

*Figure 4: Hydrograph on La_Jaillière_P4 after calibration over the entire available record, plus a close-up on hydrological year 2002-03 and the associated temporal and direct comparisons between observed (blue curve) and predicted (red curve) cumulative discharges*

During hydrological year 2002-03, the drainage season starts on the same day for both simulation and observation (i.e. November 1st). In 2002-03, the maximum observed drainage discharge lies close to 15 mm/d, versus a lower simulated peak of 12 mm/d. The SIDRA-RU model correctly predicts the temporal evolution and magnitude of drainage discharges

while accurately delimiting the drainage seasons. The simulated peak flows closely match the observed ones. Peak flows often tend to be underestimated in simulation by a few mm/d, although the timing is usually well estimated. On the whole, the dry periods are well represented by the model. Drying times are a bit longer for the simulations, but typically lie within a few days of observations. Note that spring flows are sometimes not well simulated, as was the case in 2001-02 and 2006-07.

The capacity of the SIDRA-RU model to represent the water balance has also been assessed. Figure 4 shows that the simulated cumulative drainage discharge over 2002-03 lies close to the observed discharge, with a slight underestimation of 10 mm, i.e. below 3% of the annual drained water balance relative to 350 mm. The linear regression between observed and simulated cumulative discharges yields an equation close to a 1:1 equation, with an R² of 1, leading to the assessment that the water balance is fully respected over this year on the *La_Jaillière_P4* plot, in terms of both time and total quantity.

Unlike at *La_Jaillière_P4*, the SIDRA-RU model at the *Courcival_P3* plot (see Fig. 5), which lies on a swelling clayey soil, shows larger discrepancies between observed and simulated discharges on the plots. The red and blue curves do not coincide on a significant portion of the logs. From 1985 to 1995, simulated discharges often started later than the observed ones, with delays ranging approximately from 2 weeks to 2 months. The start of the drainage season is defined here when significant discharges appear.

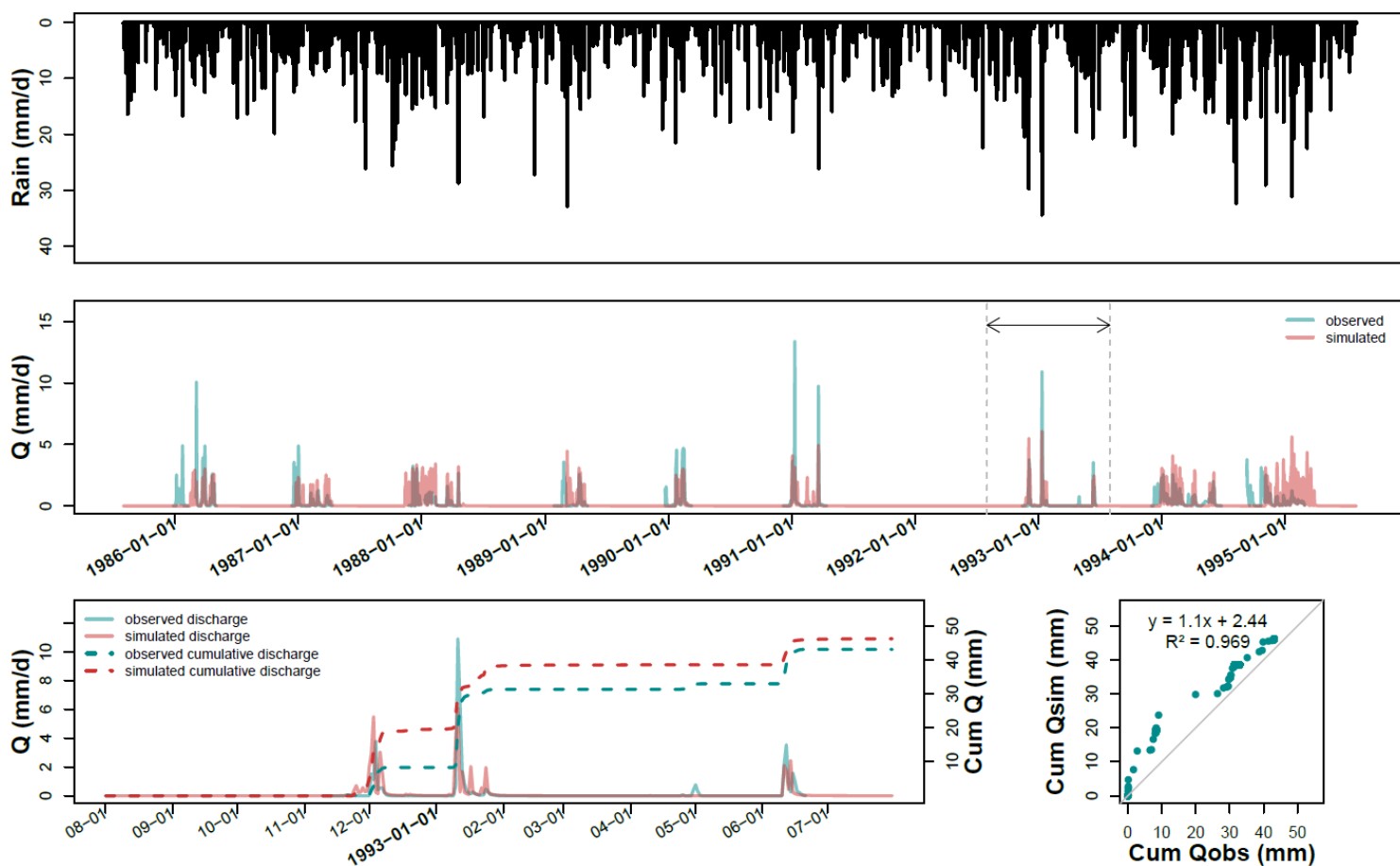

*Figure 5: Hydrograph on Courcival_P3 after calibration over the entire available record, plus a close-up on hydrological year 1992-93 and the associated temporal and direct comparisons between observed (blue curve) and predicted (red curve) cumulative discharges*

The plots for the cumulative drained discharge in 1992-93 reveal that the annual drained water balance diverges by +3 mm from simulations, but the linear regression indicates a slope equal to 1.1, which is quite high and moreover shows that the cumulative discharges have not been well simulated.

    Figure 6 provides a comparison between the predicted and observed total cumulative discharges on each plot and for each hydrological year, as classified by soil texture. The linear regressions lie close to the 1:1 equation, with an R² above 0.9

for all three textures, thereby indicating that SIDRA-RU is representing the water balance at nearly all times. However, Figure 6 does show a few discrepancies between prediction and observation, especially on the silty plots, with a deviation in the simulated cumulative discharge of 300 mm. These same observations are drawn on clayey soils, with the same discrepancies and a smaller dataset.

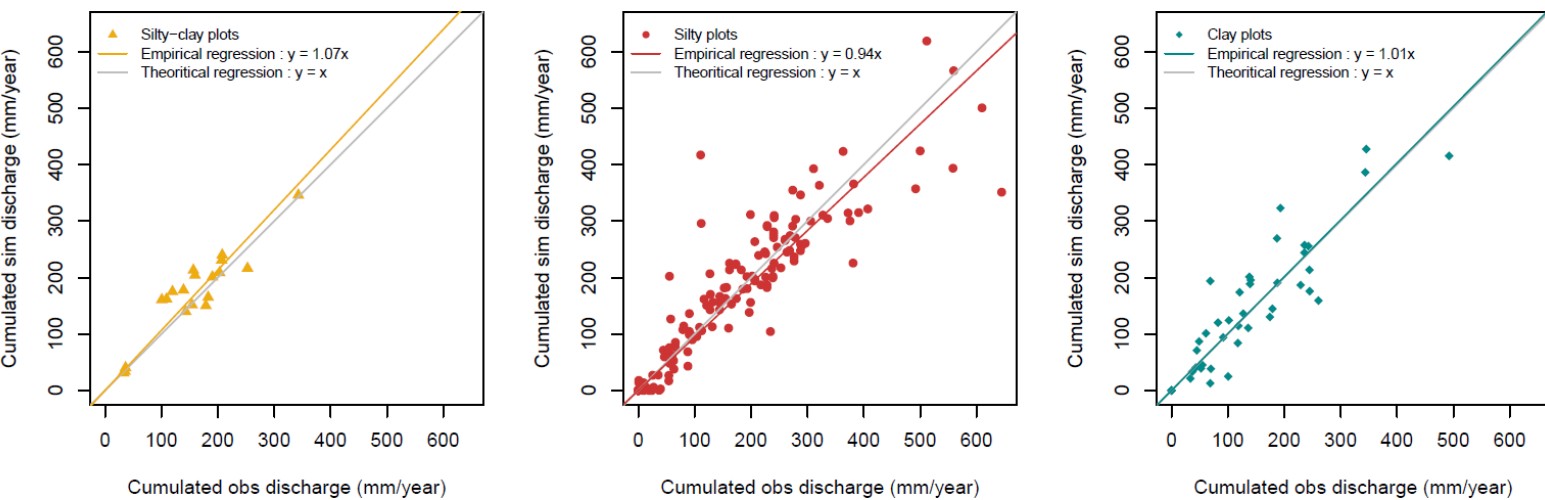

Figure 6: Comparison of the drained annual discharges between simulation and observation - each point represents the drained water quantity from a hydrological year for a given site, and all points have been classified by texture

In order to use the SIDRA-RU model for a long-term prediction of agricultural drainage management in France, the model must first be able to reproduce both high and low drainage discharges. Figure 7a depicts the differences in the Q05 quantile between observed and simulated drainage discharge. This quantile represents the values under which the annual drainage discharge occurs 5% of the time; it is used in order to evaluate low flows. The Q95 analysis (Fig. 7b) result, under which the annual drainage discharge occurs in 95% of the cases, serves to evaluate high flows. Figure 7c displays an analysis of

the average discharges ($Q_{mean}$). This $Q_{mean}$ analysis is then applied to each hydrological year of each plot on the nonzero flows, thus reducing the predicted drainage discharges by the observed ones; results are classified by texture using boxplots (Tukey, 1977). Regarding the Q05 quantiles (Fig. 7a), results show that for the three textures, bias between simulated and observed Q05 ranges from -0.020 to 0.030 mm/d, with some extreme points (mainly on the silty texture). The medians of biases all lie close to zero as well (from -0.002 to 0.002 mm/d), thus revealing that the model correctly

predicts low flows. Regarding the Q95 quantiles (Fig. 7b), the median values are once again close to zero (from -0.247 to -0.040 mm/d); however, the ranges of the Q95 biases lie above those of the Q05 quantiles. On silty soils, the boxplot limits of Q95 biases range from -1 to + 0.5 mm/d, and the whiskers range from -3 to +3 mm/d. Similarly, for silty-clayey soils, the Q95 biases vary from -3 to -2 mm/d; the discrepancies are larger on clayey soils, where the Q95 biases varies from -4 to +4 mm/d.

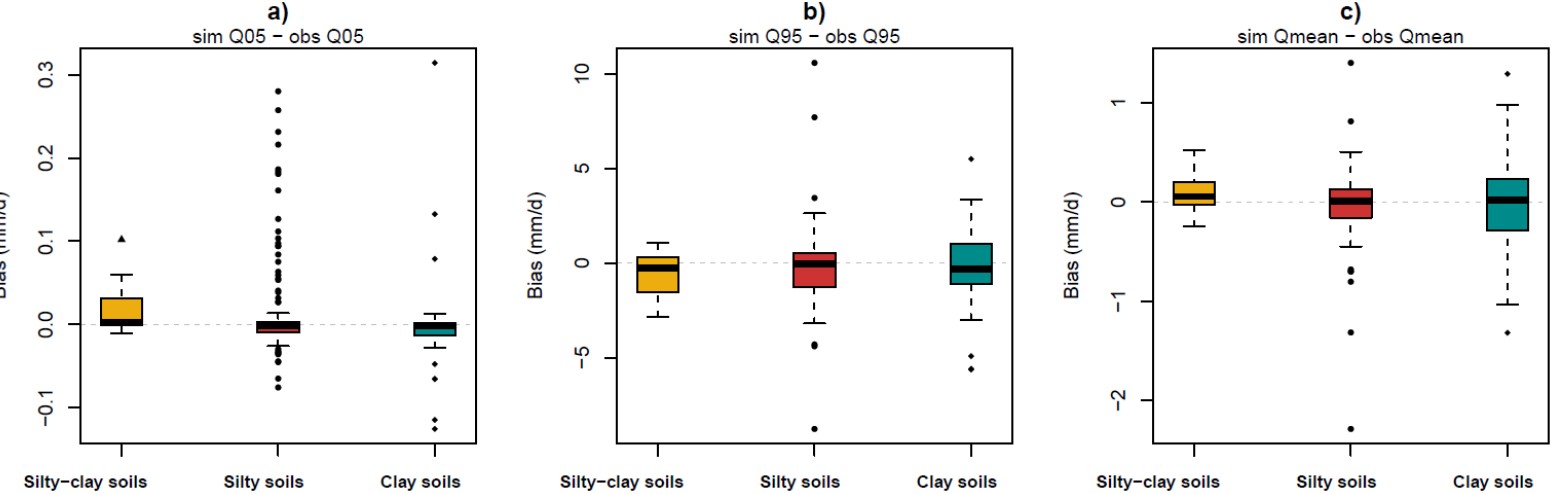

*Figure 7: Differences between prediction and observation of low (Q05), high (Q95) and average (Q_mean) flows. Each hydrological year from each site has been independently considered. Results are compiled by soil texture: 142 points for silty soils, 17 for silty-clayey soils, and 44 for clayey soils.*

Figure 7c shows that the boxplot medians for the $Q_{mean}$ biases also lie close to zero (from 0.007 to 0.057 mm/d). The $Q_{mean}$ biases range from -0.5 to +0.5 mm/d for silty soils, from -0.3 to +0.6 mm/d for silty-clayey soils, and from -0.8 to +0.9 mm/d for clayey soils. SIDRA-RU performs at a level of good agreement with respect to the average drainage discharges. The deviation on $Q_{mean}$ biases is higher on clayey soils, thus reflecting the greater difficulties of the SIDRA-RU model in simulating $Q_{mean}$ on this texture.


### 3.2 Model robustness

The KGE' values obtained during the evaluation period were then compared to those found during the calibration period, as illustrated in Fig. 8.

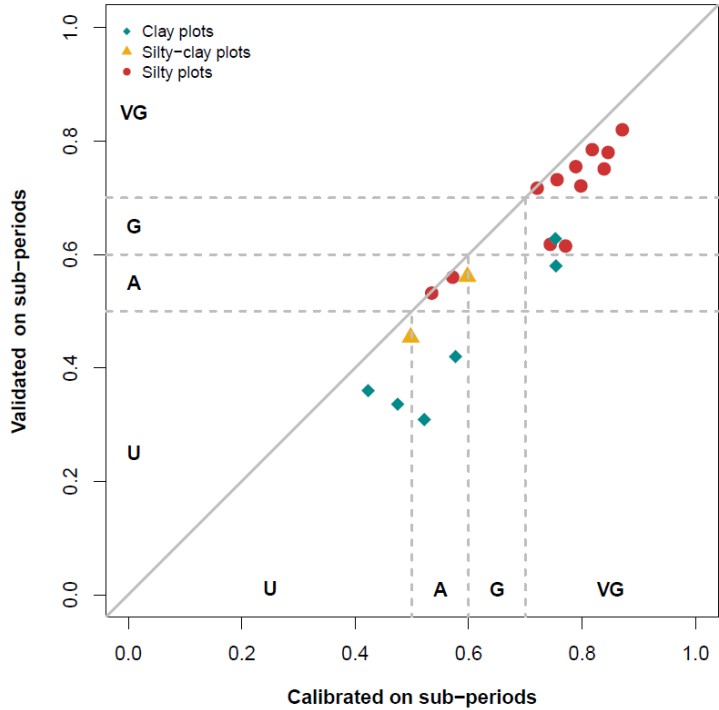

*Figure 8: Comparison between the KGE' values assessed over the calibration and evaluation periods on the 9 plots used for the split-sample test: (U) Unsatisfactory, (A) Acceptable, (G) Good, and (VG) Very Good*

For starters, this figure shows that all points are located under the line y = x, informing therefore that all evaluation KGE'
values from a specific period are always less than the calibration KGE' values from the same period. Moreover, Fig. 8
indicates that the deviations differ according to soil texture.

On silty soils, the maximum variation is observed at *Melarchez* (the largest studied site with 700 ha), with KGE' values
varying from 0.66 to 0.55 over the second sub-period (Table 1), yet the evaluation and calibration KGE' values are similar
on four of the silty plots, over both sub-periods. Furthermore, the evaluation KGE' values on silty soils remain at least
"acceptable". Results are similar for the silty-clayey soils, which feature validated KGE' values close to the calibration
values. The deviations in KGE' values are more significant on clayey soils. Indeed, KGE' varies from 0.75 to 0.58 at the
*Saint_Laurent_P2* site, i.e. a score going from "very good" to "acceptable". *Saint_Laurent_P2* is the only clayey plot that
remains at least "acceptable" according to Table 2. On the *La_Bouzule_P1* plot, the KGE' value is reduced from 0.52 in
calibration to 0.31 in validation. However, the results of Fig. 8 show that KGE' deviations on clayey soils are less than
0.21.

A graphical comparison between the predicted drainage discharges, calculated using the evaluation parameter set, and the
observed discharges helps assess model robustness, as depicted in Fig. 9 at *La_Jaillière_P4* during the 2002-03
hydrological year. The calibration simulations use the parameter set calibrated over sub-period P2 (including the 2002-
03 season), while the evaluation simulations use the parameter set calibrated over sub-period P1. Figure 9 indicates that
both calibration and evaluation drainage discharges lie close to the observed levels. The peak flows from both simulation
curves have been superimposed on the main part of the drainage season, hence the evaluation parameter set performs well
over the studied time period.

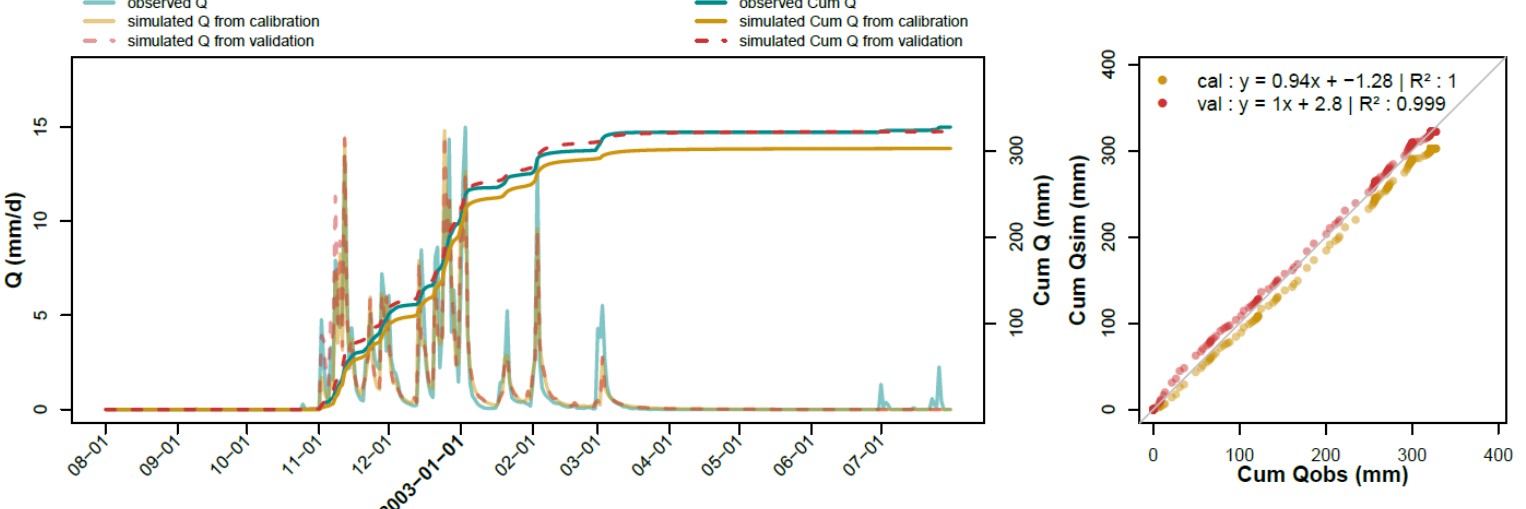

*Figure 9: Hydrographs of observed and predicted discharges on calibration (parameter set derived from sub-period P2) and evaluation (parameter set from sub-period P1) recorded at La Jaillière in 2002-03. Predicted cumulative discharges are directly compared to observations.*

Differences between the two simulations only exist at the beginning of the drainage season, as strongly controlled by the
RU module. The drainage season appears to start on the same day according to both simulation curves, but the discharge
magnitude during the first few days differs from one simulation to another. This same graphical approach has been used
on a clayey soil available in Appendix D for *Saint_Laurent_P2* during the 1982-83 season. These observations are similar
to the previous ones on *La_Jaillière_P4*.

Similar observations have also been recorded on the cumulative discharges (Fig. 9), which display an identical behavior,
except for a gap between two predicted flows appearing at the start of the drainage season and have tended to remain so

throughout the year. A direct comparison of the cumulative flows between observation and simulation yields linear regressions near the 1:1 equation, with an R² of 0.999 in evaluation and 1 in calibration. This result attests to the water balance in the calibration being close to that in evaluation. These same results are listed in Appendix D for *Saint_Laurent_P2*. With the exception of the deviation at the beginning of the curves due to an early evaluation start, the curves are parallel. The water balances are similar when using either parameter set.

The use of the split-sample test for model calibration raises the question of parameter set similarity between the two calibration periods across all sites studied. Figure 10 illustrates how each of the four parameters (K, μ, $S_{inter}$, $S_{max}$) evolves depending on the two calibration periods. Each study site has been labeled in Fig. 10 with the available corresponding index from Table 1. Results indicate that hydraulic conductivity K is the least-changing parameter between the two periods.

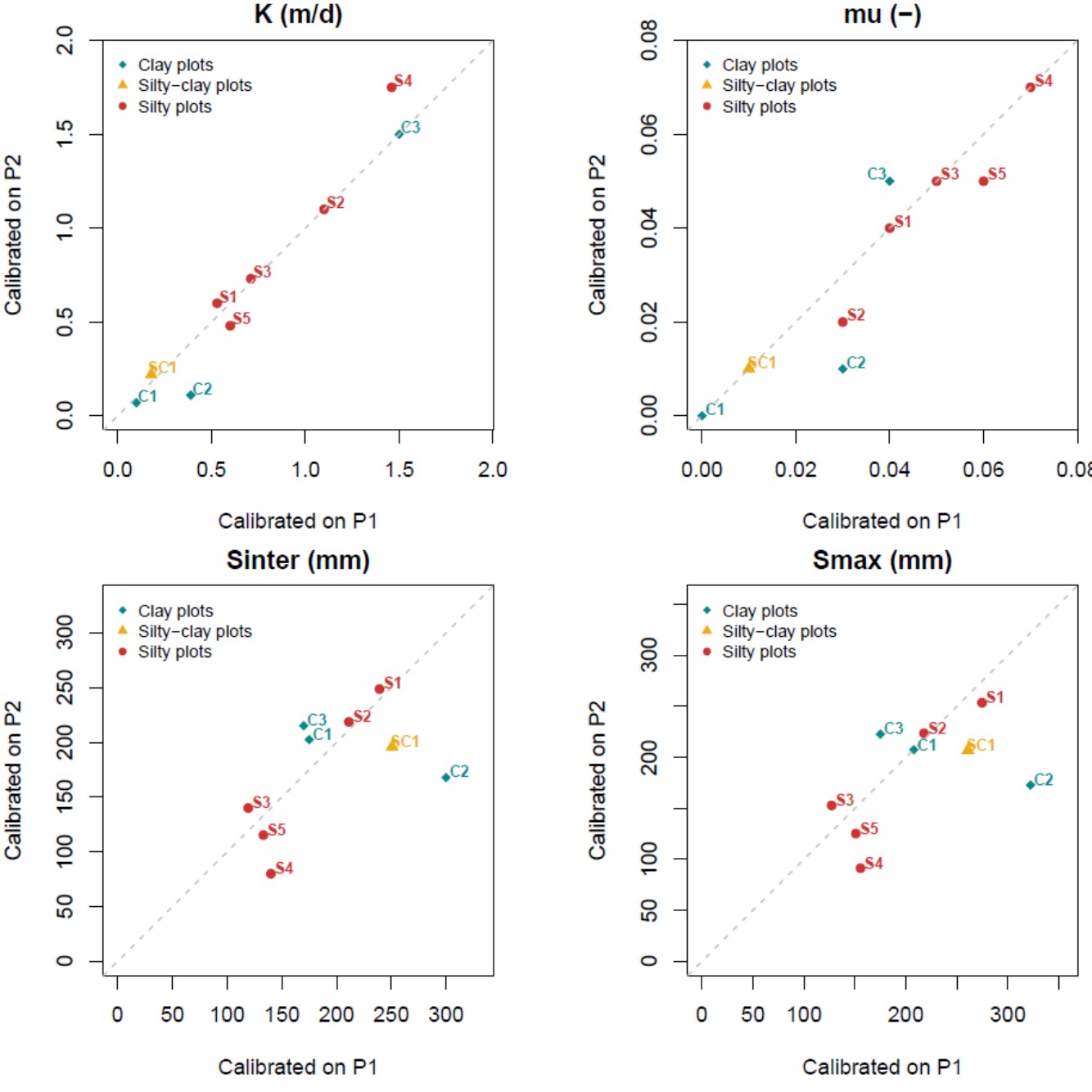

*Figure 10: Comparison of parameter sets from both calibration periods*
*(sites are referenced by their associated index, as assigned from Table1)*

One of the two extreme outlier points corresponds to *Melarchez* (Point S4), where K changes from 1.46 m/d for sub-period P1 to 1.75 m/d for sub-period P2, thus demonstrating that hydraulic conductivity values remain within the same order of magnitude. Similar results were found for drainage porosity μ, with 7 of the 9 sites lying very close to the 1:1 line, thus confirming the similarity of both calibrated μ values. *La_Bouzule_P1* (i.e. C1, Fig. 10) displays the strongest deviation, from 0.03 in P1 to 0.01 in P2, which is nonetheless a small variation and acknowledges that μ is conserved between both sub-periods.

Soil reservoir parameters $S_{inter}$ and $S_{max}$ are those exhibiting the greatest change between the two calibrated parameter sets, and in a very similar pattern. Among the 9 sites, 6 show a deviation of less than 40 mm on $S_{max}$, which seems to be acceptable. However, for the three other sites, i.e. *Melarchez* (S4), *La_Bouzule_P1* (C2) and *Saint_Laurent_P2* (C3), deviations range from 50 mm to 150 mm, indicating larger deviations for both conceptual parameters, especially on clayey soils. These observations are consistent with those derived from Figure 8 for *La_Jaillière_P4* as well as from Appendix D.

## 4. Discussion

### 4.1 A representative database

The literature (Perrin, 2000; Coron et al., 2012; Montanari et al., 2013; Thirel et al., 2015a) suggests that when assessing generalizable hydrological models, the aim typically consists of determining the ability of a model to reproduce the hydrological behaviors of various study areas. The larger the database, the more reliable the study because the model is being evaluated over a wider diversity of geological and climatic contexts (Gupta et al., 2014). To the best of our knowledge however, in drainage modeling, the models are often evaluated on a short-term database, with just a few years on a few sites, e.g. DRAINMOD (Skaggs et al., 2012), MACRO (Jarvis and Larsbo, 2012), ADAPT (Gowda et al., 2012) and SWAT (Arnold et al., 2012). This lack of data limits the opportunities to test models in various soil types and moreover prevents assessing the relevance of models over a broad array of spatial scales.

The database used in this study was specifically built to assess the performance of a drainage discharge model on a larger dataset. 22 experimental sites were compiled, accounting for nearly 200 hydrological years spread over the main drainage areas in France, on contrasted soil types and in different pedoclimatic settings featuring an extensive plot scale range. This database has been classified according to three main soil textures (silty, silty-clayey and clayey) using available drainage discharge data. Not all regions with a high drainage rate are well represented; however, 80% of French drained soils do have silty soil textures, as represented by a majority 15 of the 22 plots in the database. The real advantage of this database is the ability it offers to apply the model to referenced sites that represent a large majority of France's drainage diversity. This topic accounts for both the originality and a key contribution of this study.

### 4.2 A parsimonious hydrological model

The diversity of the database introduced requires a model that operates correctly and in accordance with each site's specific conditions, i.e. as generalizable as possible. In this context, a physically-based model is theoretically better suited. However, this kind of model is generally composed of many parameters representing the complexity of a study site such as current crop, root depth, saturated water content (i.e. unsaturated one), hydraulic parameters or water holding capacity. Performing such a model on a large database requires providing all these characteristics for each site, which is in practice

very difficult, because the measurement technics are globally expensive. The calibration might be useful but it is time consuming due to the large number of parameters. Conversely, a simple model like SIDRA-RU offers significant advantages, as it requires few information and faster calibration, thus becoming in practice suitable for generalizing. The model is lumped and, initially requiring the calibration of six parameters, the assumptions made in this study allow to reduce this number to four parameters. Compared to the original SIDRA model (Lesaffre and Zimmer, 1987a), the current version simulates continuous discharges over several hydrological years, thus simulating both wet and dry periods. The calibration process only necessitates brief rudimentary knowledge of the soil texture, used here as *a priori* input data to establish the model parameter distributions. Regarding input variables, the model requires rainfall and PET data at a daily time step to predict the drainage discharge. Managed in conjunction with the SAFRAN climate database to satisfy data needs across France, SIDRA-RU can easily be launched on all drained areas throughout the territory.

### 4.3 SIDRA-RU model performance

Once calibrated, the model shows a performance ranging from "acceptable" to "very good" on all sites, judging from the KGE' value, except for clayey soil. The model provides accurate simulations on small plots (0.8 ha at *Parisot_P2*) as well as on large plots (700 ha at *Melarchez*). In comparing the SIDRA-RU performance with that of other models tested in more local studies (Gowda et al., 2012; Skaggs et al., 2012; Muma et al., 2017; Revuelta-Acosta et al., 2021), SIDRA-RU exhibits an equivalent simulation quality. For a vast majority of simulations carried out by the model, the observed annual water balance is indeed reproduced with a good agreement. These performances are close to those obtained using pure physical approaches, like the models RZWQM (Ma et al., 2007) and ADAPT (Sands et al., 2003). Moreover, this congruence in annual cumulative water quantity between observation (integrating drained discharge and runoff) and simulation (without runoff) validates the assumption that considers runoff to be negligible, according to the conclusions of Kuzmanovski et al. (2015). Similarly, this result validates the assumption made regarding the fact that a large majority of drained soils lie on an impervious layer, hence authorizing us to neglect the recharge to groundwater. This result also supports the assumption made regarding the fact that the effect of the current crop growth is neglected as long as subsurface drainage occurs mainly in winter on non-fallow soil. Also, the SIDRA-RU model allows simulating the temporal variation of drainage discharge for both dry and wet periods; moreover, flood peaks are simulated on time as are recession periods, thus offering a performance comparable to that of more complex models, e.g. DRAINMOD (Skaggs et al., 2012), CATHY (Muma et al., 2017) and RZQWM (Jiang et al., 2020b). Overall, the simple design of the SIDRA-RU model allows achieving a performance at least as good as that found in current models, like MACRO on the *La Jaillière* plot (Kuzmanovski et al., 2015) or WEPP (Revuelta-Acosta et al., 2021).

Let's note however that some peak flows have been slightly underestimated. This behavior is generalizable, being observed on most database sites. However, the phenomenon varies from one site to another without any clear relationship with any soil characteristic, mostly depending on the calibration quality. One explanation might be that the SIDRA-RU model is calibrated on both high and low drainage discharges, as represented by a unique parameter set, which consequently introduces biases. These biases become more significant on extreme discharges. Furthermore, the missed variations sometimes lead to missing the beginning of the drainage season. During some hydrological years, the observed drainage discharge starts before water level in the soil reservoir reaches the $S_{inter}$ threshold during simulation. Indeed, a rainfall event appearing from August to October is not specifically converted to discharge every time, mainly due to the nonlinear processes controlling the precipitation-discharge transformation. Furthermore, the assumptions related to

SIDRA-RU, e.g. neglecting lateral communication with other plots of land, might also be responsible for this phenomenon. This early start might prevent completely filling the annual water balance. However, deviations in this indicator remain minor in the large majority of the cases, i.e. from 5% to 10%, and produce no significant consequences on long-term studies.

Regarding the split-sample test, results first showed that evaluation KGE' values were all lower than calibration KGE' values from the same period. This finding seems to be normal and assesses the consistency of the test since an evaluation parameter set will normally always yield a lower KGE' value than the one obtained from the calibration set, thus corresponding to the best result depending on the chosen OF (here the KGE' criterion). Second, performances using calibration parameters from both sub-periods are similar across all sites, with the evaluation KGE' values being close to the calibration values and moreover remaining quite good, especially on silty and silty-clayey soils. Performance is lower on the clayey soils, while still remaining acceptable when considering the difficulties the model experiences on clayey soils during calibration. The drainage discharge behaviors are similar as well, as logs from both evaluation and calibration simulations do merge on most occasions with equal water balances. These results attest to the temporal robustness of the SIDRA-RU model on the studied textures, which represent the largest proportion of drained areas in France.

## 4.4 Calibration consistency

Another important analysis is the consistency of model calibration. The SIDRA module actually solves a simplified formula derived from the Boussinesq equation and requires a good estimation of both hydraulic parameters, namely hydraulic conductivity K and drainage porosity $\mu$. Thus, model calibration is only reliable if the calibrated K and $\mu$ are probable according to the case study soil type. Figure 11 compares the distribution of the calibrated values of silty plots with the distribution obtained using the measured values of K and $\mu$ during tests conducted on reference silty drainage sites. The theoretical curves have also been drawn according to these distributions using the mean and standard deviation of each sample while following a log-normal law. The database provided only allows assessing the quality of the calibration on silty soils since 15 sites are referenced for this particular texture. On both of the other soil textures, only three sites are available for silty-clayey soils and four for clayey soils, which is insufficient to compare the calibrated values to a reference base, hence constituting a limitation of this study.

Regarding hydraulic conductivity K, some divergences exist between the reference base and the calibrated values, especially for the highest hydraulic conductivity values, i.e. close to 2 m/d. However, over the remaining range of values, the histogram from the calibrated K are congruent with the one from the reference drainage tests. The theoretical laws are also similar between the reference base and calibrated values even though the curve derived from the calibrated value shows a higher standard deviation, due to high hydraulic conductivity values. Despite some discrepancies, the distribution of calibrated K values for silty soils seems to be reliable on the basis of soil type. As for $\mu$, the same method has been applied (Fig. 11); the histograms and theoretical laws do match, thereby concluding that the calibration process reliably estimates $\mu$ as well. The split-sample test confirms this analysis, in showing that K and $\mu$ from silty sites are conserved from both calibration sub-periods. Each K and $\mu$ value seems to be calibrated with a robust and consistent data point, attesting to the relevance of calibration on the physically-based module.

As regards $S_{inter}$ and $S_{max}$, these parameters are conceptual and no observed data can be used as a reference to define their statistical distributions. Furthermore, even if they were to constitute a conceptual approach of water holding capacity, soil texture is insufficient as a description, and no distributions can be determined based on the database classification employed. One solution might consist of comparing the $S_{inter}$ and $S_{max}$ of each site with the associated actual value of the water holding capacity; however, the database does not provide this information for every experimental site and measuring it *in situ* would be very expensive, thus infeasible. Nevertheless, it can be noticed that the range of values of $S_{max}$ are consistent with water holding capacities of those kinds of soils ranged from 70 to 200 mm for 1 m depth of soil.

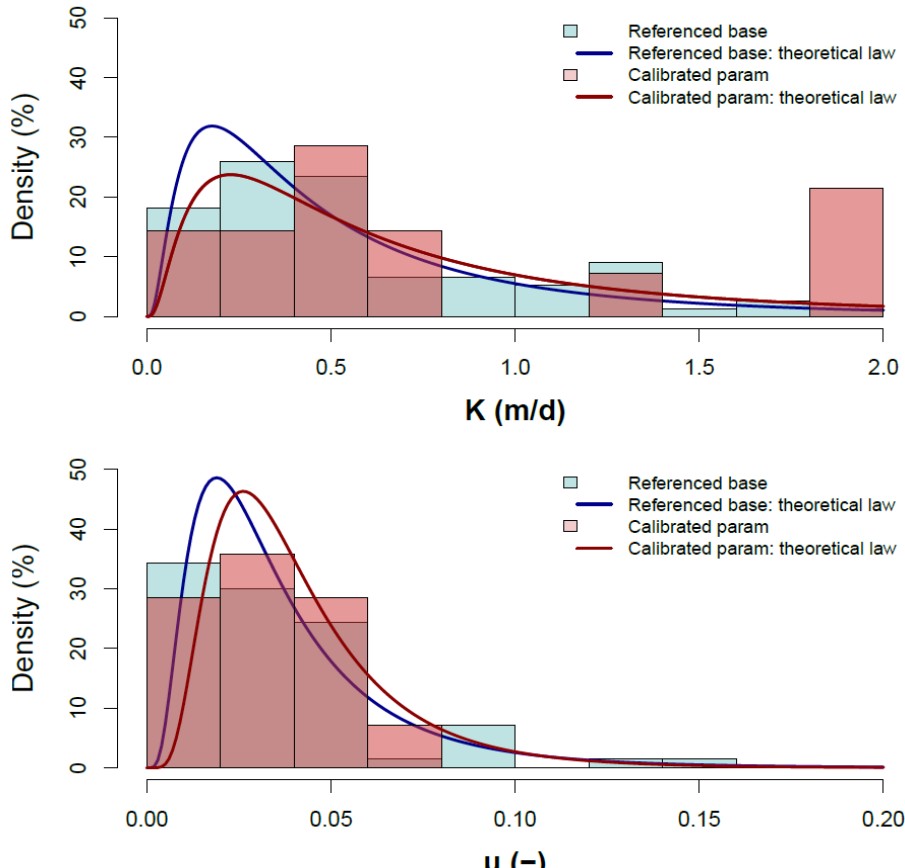

*Figure 11: Distribution of K and µ from calibrated values on silty soils vs. values extracted from reference drainage sites placed on silty soils*

## 4.5 Weaker performance on clayey soils

On non-deformable clayey soils, such as *Saint_Laurent_P2*, SIDRA-RU produces a good performance. However, at *Courcival_P3*, a plot lying on a deformable swelling clayey soil, the SIDRA-RU performance is significantly poorer. As such, the latter does not constitute an exception in the drainage modeling community. Indeed, the literature identifies clayey soils as a recurring problem for drainage modeling (Robinson et al., 1987; Snow et al., 2007), especially in mole drainage, as currently practiced on heavy clayey soils and swelling clays (Jarvis and Leeds-Harrison, 1987; Tuohy et al., 2016). This finding is mainly due to a difference in hydraulic characteristics between silty soils, on which the model design is primarily based, and heavy or swelling clayey soils. The latter are characterized by natural pedological deformations, like soil surface fracturing, which lead to preferential flow zones before saturation (Beven and Germann, 1982; Jarvis and Leeds-Harrison, 1987). The horizontal soil profile is no longer homogeneous, which contradicts one of the main hypotheses of SIDRA. Moreover, agricultural practices like plowing exacerbate this phenomenon and therefore

affect soil porosity. One way to improve results at *Courcival_P3* would be to artificially locate the pipe at a depth of 30 cm instead of the current 90 cm.

Another critical assumption of the SIDRA module is the elliptical shape of the water table; this assumption facilitates the numerical resolution of the Boussinesq equation. As regards heavy clayey soils, this hypothesis is no longer suitable since the water table shape evolves towards a rectangular structure (Fig. 12). This phenomenon is due to the very low hydraulic conductivity (Robinson and Rycroft, 1999; Skaggs et al., 1999), which for SIDRA-RU is difficult to integrate. Furthermore, the water table level drops when approaching the drain because the soil has been turned over on this profile in order to bury the drain, like for the aforementioned natural pedological deformations.

Furthermore, the issues observed on clayey soils are specifically significant at the start of the drainage season. At *Saint_Laurent_P2*, delays occur more frequently than on silty soil, so the more the plot is defined by heavy or swelling clayey soil, the longer the delay. At *Courcival_P3*, these delays were on the order of one month. The RU module design partially addresses this problem, by generating a soil profile recharge before saturation, but this issue remains a major limitation of the model, i.e. the more clayey the soil, the poorer the model performs. A soil is considered to be mainly clayey once the clay fraction exceeds 35% (Richer-de-Forges et al., 2008), which can be defined as the limit beyond which good model performance is no longer guaranteed.

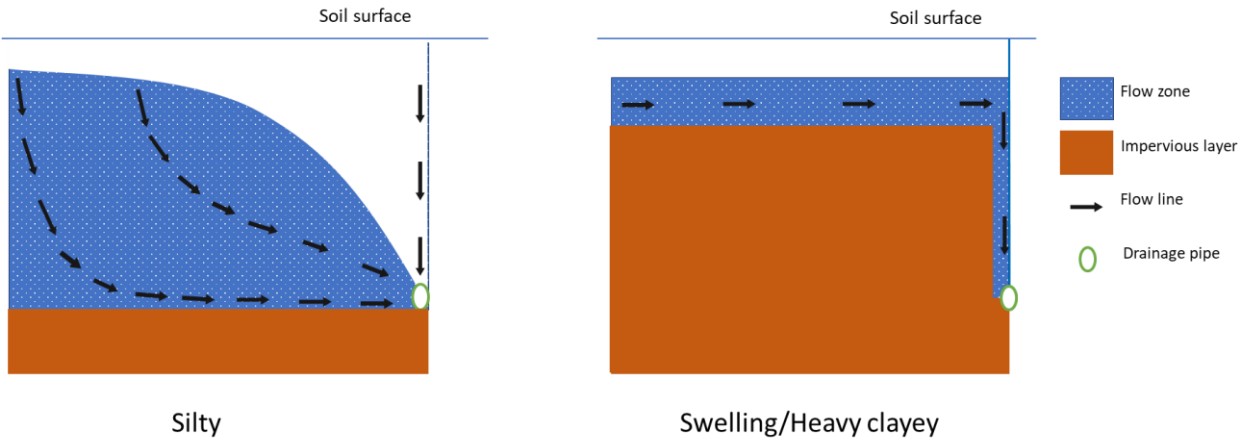

*Figure 12 : Evolution of the water table shape between a silty and a heavy clayey drained soil, adapted from Bouarfa and Zimmer, 2000 and Branger et al., 2009*

Some models show their efficiency when simulating drainage discharge on such soils. Among them, there is the MACRO model which showed its efficiency to simulate drainage discharge in Europe, including structured soils like heavy clayey soils (Köhne et al., 2009). However, these models are based on a more physically-oriented approach than SIDRA-RU. Building on this type of concept would probably improve the latter in simulating drainage discharges on clayey soils, but might undermine its generalist nature.

**4.6 A less robust RU module**

Another limitation of this model is the RU parameters are slightly less temporally robust than the SIDRA parameters, showing fewer stable values between calibration sub-periods than the SIDRA parameters. This issue might originate from dependencies of the RU module on interannual meteorological variations. The $S_{inter}$ parameter represents the soil storage threshold that allows the model to initiate each drainage season. Compared to a common hydrological period without too much extreme discharge events, a dry calibration period, defined by a larger occurrence of dry events on a same period due to climate conditions, increases $S_{inter}$ values hence the model stores a larger quantity of water, delaying the start of the drainage season. Conversely, a wet calibration period will tend to decrease $S_{inter}$ in anticipation of the start of the

drainage season. Regarding *Courcival_P3*, the split-sample test showed that $S_{inter}$ evolved from 175 mm on 1986-1990 to 202 mm on 1990-1995, a dry period (see Fig. 5). Consequently, if the two sub-periods are meteorologically contrasted, $S_{inter}$ (so as $S_{max}$) will differ from the two calibrated parameter sets. One solution might be to consider the $S_{inter}$ parameter as a variable to be adjusted according to the meteorological conditions of the previous year as well as to soil parameters, e.g. whereby the actual water holding capacity is defined as a new parameter. Clayey soils intensify this robustness issue due to weather fluctuations exacerbating the formation of preferential flow zones.

### 4.7 Interpretations bounded by choices

Another point we wish to discuss deals with the dependency of the aforementioned interpretations on the conditions and choices involved in conducting this study. We noted above the lack of data in the database used for certain French regions. Making assumptions, e.g. grouping soils by categories, to address this weaknesses would then offer a good alternative, yet still dependent on arbitrary decisions. Adding reference sites could complete and significantly improve the SIDRA-RU model robustness analysis.

Moreover, the model is based on some rather important assumptions, such as neglecting both the surface runoff and recharge to groundwater, or the ones made on the parameters $a$ and $\alpha$, which limit its field of application. Indeed, even if these assumptions do seem to be acceptable on sites used to represent French subsurface drainage, up to now there is no evidence of their relevance on other sites. This limits the use of SIDRA-RU to the specific conditions outlined herein. Nonetheless, it should be noted that SIDRA-RU may be used without this assumption, in integrating the depth infiltration with Hooghoudt's equation, according to the principle of equivalent depth (Zimmer, 1992). A term "$D_s$", designating the deep seepage rate is introduced in the Boussinesq's equation to reduce the recharge rate to the drains.

Furthermore, the choice of calibration process drives the calibration results based on study goals. We have implemented herein a grid-screening algorithm that assigns the best combination of parameters according to their respective distributions, coupled with a step-by-step algorithm. This approach has the advantage of being theoretically entirely automatic and thus eliminates the subjective aspect of calibration; however, external decisions influencing results are still necessary.

As an example, in order to track the main purpose of this study, SIDRA-RU is calibrated using the KGE' criterion as its OF, in combining three criteria (Kling et al., 2012), for a relevant approach to properly representing the interannual variability of drainage discharge. However, if this model is to be used for another purpose, like predicting drainage season initiation, then the KGE' criterion might not be the most efficient OF. This statement highlights the fact that the SIDRA-RU model is robust with respect to KGE', yet nothing proves the same for other OFs or purposes.

The distribution functions of each model parameter are required for the calibration, which in this case depends on the decision to classify soils based on their texture. Regarding K and µ, these distributions are established from measured data extracted from the reference drainage tests conducted in the 1980's, as mentioned above. These measurements are subjected to uncertainty, depending on both the number of measurements and the method employed; they constitute a significant source of error. We also mentioned above that classifying a soil by its texture alone is a major assumption and one that distorts the distributions. Consequently, the calibrated parameters, constraint by the distributions a priori defined, might be biased, due to an overly wide range of referenced values that bias the mean and standard deviation, as shown in Fig. 10 on silty sites. This problem is particularly worrisome since both those parameters are physical, making it prohibited to set outliers.

Driving the distribution with realistic value ranges at each site can prove to be a relevant solution. For example, if we consider that on a specific soil K might be included within a smaller value range, according to information obtained from the study site, then the calibration might lead to a more realistic value of K. This strategy reduces the risk of extracting parameters with secondary optima, and moreover calibration accuracy is refined. However, better knowledge of the soil characteristics is required, with the assumption that by knowing the reliable ranges, calibration becomes semi-automatic, 610     thus introducing an arbitrary decision-making factor into the process. In addition, this supplementary knowledge requires relatively complete databases of drained plots, which as previously discussed constitutes a persistent issue.

### 4.8 Coupling with modules to simulate pollutant leaching

    In the perspective of long-term management on drained plots, predicting flows in order to better monitor the use of 615     agricultural pollutants is a major concern, pollutant transfers occurring with drainage flow (Kladivko et al., 2001; Trajanov et al., 2018). Thus, a good model can be used as a decision-making tool, for example to restrict pollutants' application during flow period for the case of pesticides (Lewan et al., 2009; Zajíček et al., 2018; Kobierska et al., 2020). Indeed, using water content as an indicator to anticipate the start of drainage flow in order to reduce pesticide applications is a recommended strategy instead of restriction timing (Brown and van Beinum, 2009; Lewan et al., 2009). In this context, 620     using SIDRA-RU may be quite relevant. However, the current form of the RU module is not optimal to accurately represent the fate of pollutant in soil profile, being too simple to precisely represent the behaviour of the water table inside the unsaturated zone. To overcome this problem, this model type is generally coupled with pedotransfer functions (Jury and Roth, 1992; Magesan et al., 1994) to transfer water and pollutant stock from the unsaturated zone to the saturated zone. Within this framework, the perspective of the PESTDRAIN module (Branger et al., 2009), coupled with the SIDRA-625     RU model, should allow simulating pesticide leaching by including two reservoirs: fast reservoir to mimic preferential flow above the drain area and slow reservoir through the soil matrix. Based on a similar approach, combining SIDRA-RU with a nitrate leaching module might also be useful in order to correctly assess water pollution on French drained plots.

630

### 5. Conclusion

    The aim of this study has been to implement an large database characterizing the main drainage areas throughout France, so as to assess the performance and robustness of the SIDRA-RU model. A database comprising 22 drained sites was built to represent French soil diversity and describe the three main soil textures from France's main drained parts (i.e. 635     silty, silty-clayey and clayey). Results indicate that the SIDRA-RU model yields satisfactory drainage discharge simulations for nearly all studied sites. Moreover, the model shows especially good performance on silty soils, which account for 80% of all drained plots in France. Despite a number of limitations, particularly for clayey soils, the model was found to be temporally robust at the national scale, which enables conducting long-term impact studies. Once calibrated, this model can indeed be used to assess the resilience of drainage systems under climate change according to 640     climate scenarios like those from the CMIP5 project (Eyring et al., 2007; Taylor et al., 2011) or the upcoming CMIP6 project.

## 6. Appendices

| Texture | K (m/d) | | μ (-) | | Sinter (mm) | | Smax (mm) | |
|---|---|---|---|---|---|---|---|---|
| | **Mean** | **SD** | **Mean** | **SD** | **Mean** | **SD** | **Mean** | **SD** |
| **Clayey** | 0.32 | 2.53 | 0.017 | 2.081 | 138.39 | 53.33 | 171.68 | 70.19 |
| **Silty-Clayey** | 0.99 | 3.52 | 0.018 | 2.189 | 138.39 | 53.33 | 171.68 | 70.19 |
| **Silty** | 0.90 | 3.17 | 0.031 | 1.926 | 138.39 | 53.33 | 171.68 | 70.19 |

*Appendix A: Table of mean and standard deviation (SD) per soil texture used for the parameter distribution*

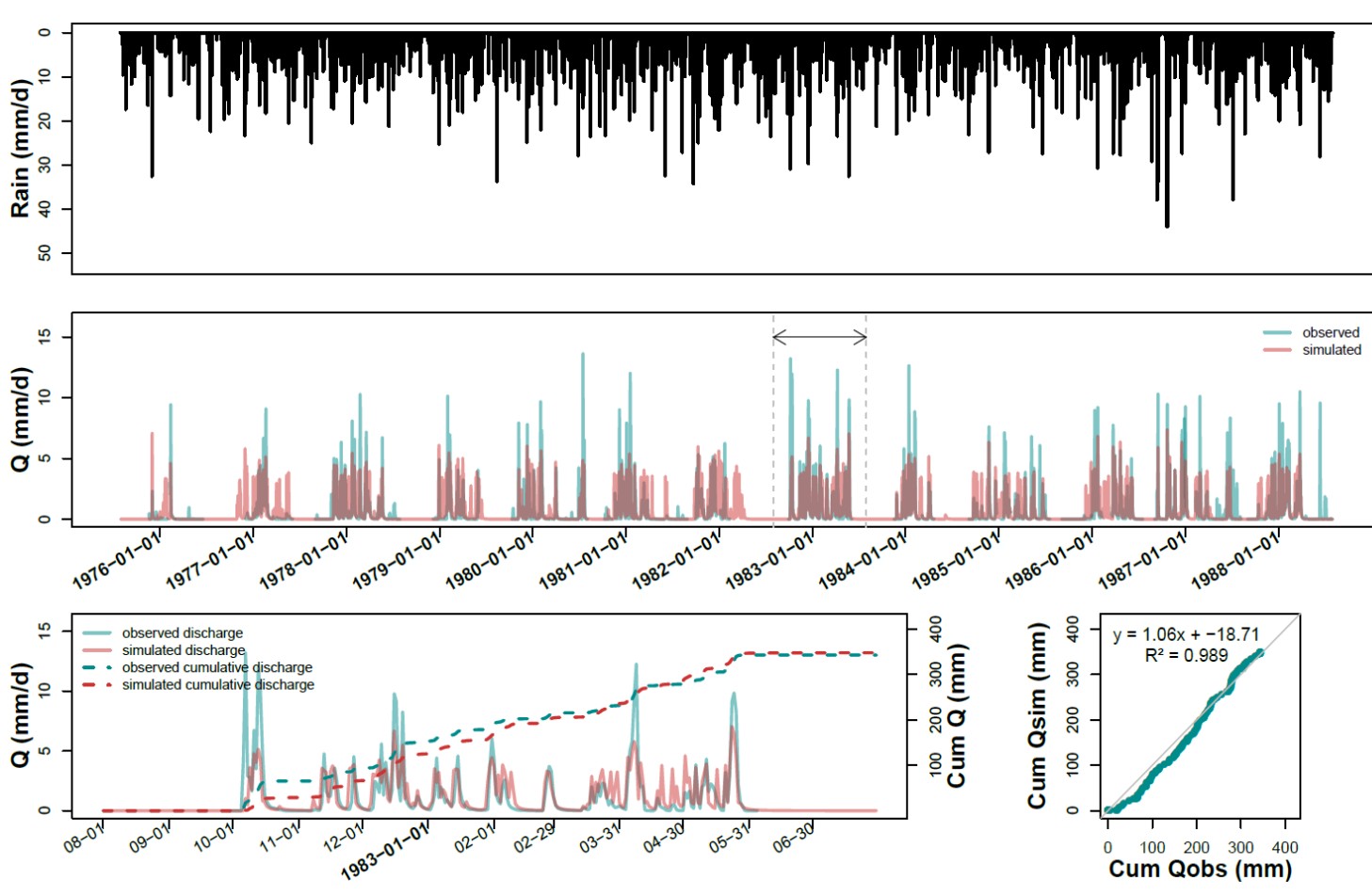

*Appendix B: Hydrograph on La_Bouzule_P2 after calibration over the entire available record, plus a close-up on hydrological year 1982-83 and the associated temporal and direct comparisons between observed (blue curve) and predicted (red curve) cumulative discharges*

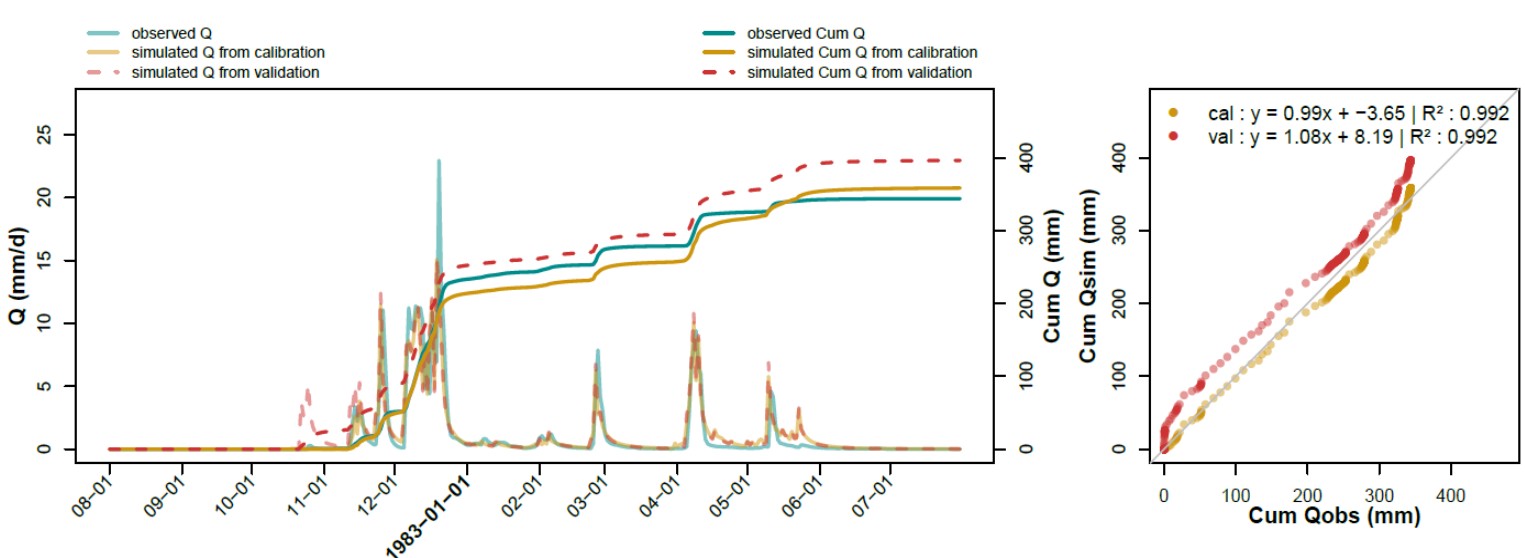

*Appendix C: Hydrograph on Saint_Laurent_P2 after calibration over the entire available record, plus a close-up on hydrological year 1982-83 and the associated temporal and direct comparisons between observed (blue curve) and predicted (red curve) cumulative discharges*

645

*Appendix D: Hydrographs of observed and predicted discharges on calibration (parameter set derived from sub-period P2) and evaluation (parameter set from sub-period P1) recorded at Saint_Laurent_P2 in 1982-83. Predicted cumulative discharges are directly compared to observations.*

*Author's contributions*. AJ, HH and JT conceptualized the work. LC and GT provided methodological guidelines. CC, HH and JT collected data and rendered them sustainable. AJ drafted the paper. HH, CC, LC, GT and JT all revised the paper and contributed to its analyses and discussions.

*Competing interests*. The authors hereby declare that they have no conflict of interest as regards the present work.

*Acknowledgments*. This work benefited from the French state aid managed by the ANR under the "Investissements d'avenir" programme with the reference ANR-16-CONV-0003 (CLAND project). Our gratitude is also expressed to the Météo-France Meteorological Agency for providing the data used in this work from the SAFRAN database. Moreover, thanks are addressed to both the ARVALIS Plant Sciences Institute for data from the *La Jaillière* measurement site and the ORACLE Research Observatory for its data. The authors would like to thank the editor (Dr C. Stamm) and the two reviewers for their constructive and kind comments and suggestions.

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
