# Peer review of "Robustness of a parsimonious subsurface drainage model at the French national scale"

_Hydrology and Earth System Sciences, 2021_

## Author Comment (AC1)

**Reply to Nicholas Jarvis**

We would like to thank Nicholas Jarvis for his truly relevant comments on the manuscript and we are glad that he finds that this work may show an interest for the hydrological community of subsurface drainage. We agree with most of the comments and we propose an answer for each specific comment and a suggestion for correction accordingly hereafter.

Comment:

*1. Line 177: The origin and meaning of this equation is unclear and should be explained. Perhaps a figure could help? Where does the number 0.4 come from? In principle, this looks like another parameter to me, even if you assumed it here to be a constant.*

- Reply: the reviewer is right, this point remained unclear in the text. The idea is to consider the volume of the storage from 0 to the $S_{inter}$ parameter as an approximate concept of the water holding capacity. As so, the available water capacity that could be easily used by crops is a part "$a$" of this volume. We agree with the reviewer, $a$ might be considered as a fifth parameter to be calibrated. However, to keep SIDRA-RU as parsimonious as possible, $a$ is set at a fix value. Indeed, Tournebize et al. (2015) showed that $a$ might be included from 60% to 70% of the water holding capacity on French drained plots. Thus, we choose to fix this proportion at 60% and so to fix the $S_{RFU}$ level at 40% of $S_{inter}$. We propose the following correction at the line 176 of the original version of the manuscript after "Two parameters control the RU module.":

"Two parameters control the RU module. On the one hand, the $S_{inter}$(mm) parameter is an intermediate threshold of the soil reservoir defining the water quantity needed to generate flow in the reservoir before saturation of the storage (see Eq. (2)):

$$S_{RFU} = a * S_{inter} \tag{2}$$

The factor $a$ is set to 0.4 due to the water capacity that could be easily usable by the crops (RFU for "Réserve Facilement Utilisable" in French) representing approximately 60% of $S_{inter}$ (approximate concept of the water holding capacity) on French drained soils (Tournebize et al., 2015)."

Comment:

*2. Line 186: in principle, α is also a parameter that apparently has been previously calibrated against experimental data and is now set as a constant. However, in principle, it should depend on soil hydraulic properties and is therefore not a constant for different soil types. It's not a very important point, but the author's claim that there are only 4 parameters to calibrate in this model is in my opinion a little dubious. As far as I can see, it should be six.*

- Reply: the reviewer is right, α might also require a calibration. However, a sensitivity analysis from an article currently being published (Henine et al., In publication) showed that α is not sensitive to the KGE' criterion used in the study. Consequently, to limit uncertainties relative to the calibration process for a non-sensitive parameter, α is set to $1/3$ due to previous experimental tests.

Comment:

*3. Line 202: You should also mention another important assumption here. You also assume that recharge to groundwater is negligible i.e. that all excess water is routed to the drainage system. This seems to be a reasonable assumption for your study sites, because the water balances are simulated quite well, but it will definitely not always be the case. For example, wide-spaced drainage systems are often installed in fields with slowly permeable subsoils (e.g. in soils with morainic parent material). Annual recharge to groundwater is definitely not a negligible term of the water balance in such cases. It should be discussed here whether your model could be adapted to account for this kind of hydrogeological situation (and if so, how).*

- Reply: The referee is right, recharge to groundwater is neglected in this study. We will mention this assumption in the revised version of the manuscript. French drained plots suffering from permanent infiltration issues represent less than 10% of the total drained areas (Lesaffre, 1987) and since 2009 the French law prohibits the use of drainage systems on such soils. Furthermore, most of French drained soils belongs to the hydromorphic soils category showing temporary waterlogging issues due to a perched water table (Lagacherie and Favrot, 1987). In this context, neglecting recharge to groundwater seems to be quite reasonable. However, SIDRA-RU may be used without this assumption, integrating the depth infiltration to the Hooghoudt's equation according to the principle of equivalent depth (Bouarfa and Zimmer, 1998).

-

Comments:

*4. Lines 301-303: It would help readers to better understand the model limitations for clay soils if the authors also showed some plots of the results for the Courcival site at this point.*

- Reply: the reviewer is right, presenting the case of Courcival, among the clayey soils, would obviously be interesting to show the reader a situation in which SIDRA-RU is not good, introducing a new figure. We propose the following correction after the line 314 in the Results part:

"Unlike at *La Jaillière*, SIDRA-RU at *Courcival* (see Fig. 5), plot lying on a swelling clayey soil, shows larger discrepancies between observed and simulated discharges on the plots. The red and blue curves do not coincide on a significant part of the chronicles. From 1985 to 1995, simulated discharges often start later than the observed ones with delays approximately ranging from 2 weeks to 2 months. The start of the drainage season is defined here when significant discharges appear. The plots on the cumulative drained discharge on 1992-93 show that the annual drained water balance diverges by +3 mm from simulations but the linear regression shows a slope equal to 1.1, which is quite high and shows that the cumulative discharges are not well simulated. These results are consistent with the fact that SIDRA-RU at *Courcival* shows an unsatisfactory KGE' value."

[Figure]

"

We also propose to reword the paragraph from line 498 to line 507:

"On non-deformable clayey soils such as *Saint Laurent*, SIDRA-RU shows relevant performances. However, at *Courcival*, plot lying on a deformable swelling clayey soil, SIDRA-RU performances are significantly weaker. In this respect, the latter is not an exception in the drainage model community. Indeed, the literature identifies clayey soils as a recurring problem for drainage modeling (Robinson et al., 1987; Snow et al., 2007), especially in mole drainage, as currently practiced on heavy clayey soils and swelling clays (Jarvis and Leeds-Harrison, 1987; Tuohy et al., 2016). This finding is mainly due to a difference in hydraulic characteristics between silty soils, on which the model design is primarily based, and heavy or swelling clayey soils. The latter are characterized by natural pedological deformations, like soil surface fracturing, that lead to preferential flow zones before saturation (Beven and Germann, 1982; Jarvis and Leeds-Harrison, 1987). The horizontal soil profile is no longer homogeneous, which contradicts one of the main hypotheses of SIDRA. Moreover, agricultural practices like plowing exacerbate this phenomenon and therefore affect soil porosity. One way to improve results at *Courcival* is to artificially locate pipe at 30 cm depth instead of 90 cm as it is."

Comment:

*5. Lines 310-311: do you mean that the model gets the timing of the start of drainage in clay soils wrong by a whole month? Please clarify. It would help to show this in a figure (see point 4).*

- Reply: Indeed, the start of the drainage season, i.e. the date when the first significant discharges are observed, on clayey soils often appears with one-month delay in simulation compared to observed data. Without being systematic, this is mainly specific to this soil texture, the delay is weaker on silty soils and silty-clayey soils. As you propose, the latter figure may also be used to illustrate that half of the studied period is concerned by this delay. We propose to include this explanation to the correction proposed for the comment 4. We propose to add the following correction at the line 508 after the above-mentioned paragraph proposed as correction:

"Furthermore, the issues observed on clayey soils are specifically significant on the start of the drainage season. At *Saint Laurent*, delays occur more frequently than on silty soil and the more the plot is defined by heavy or swelling clayey soil, the larger the delay. At *Courcival*, these delays are around one month."

Comment:

*6. Lines 331-332 and 579-581: it will not be possible to accurately model pesticide or nitrate leaching based on the RU hydrological model, because the unsaturated zone is modelled as a single box. Leaching of nutrients and contaminants in soil cannot be accurately modelled using such a lumped hydrological model for the unsaturated zone. Physics-based modelling approaches are necessary. This should be acknowledged in some suitable way.*

- Reply: the authors do not completely agree with this comment. The referee is right, it is often necessary to use more physically based approaches than the RU module to represent the unsaturated zone to accurately model pesticide and nitrate leaching. However, using pedotransfer functions from conceptual approaches is also possible (Jury and Roth, 1992), and already used on drained soils (Magesan et al., 1994). Combined with SIDRA-RU and within this framework, the PESTDRAIN module already exists for the pesticide leaching (Branger et al., 2009) and the SIDRA-N module is currently developed to simulate nitrate leaching in the team to which the main author of the present study belongs. Both of modules include several compartments (one fast and one slow pathways) with few parameters and show interesting performances.

Comment:

*7. Lines 498-500: Yes, most hydrological models work less well on clay soils and SIDRA-RU does not seem to be an exception. But it should be acknowledged that there are exceptions. For example, MACRO was originally designed to simulate flow processes in structured soils and it has been shown to work very well in heavy clays (Köhne et al., 2009). There are three main reasons for this. It accounts for non-equilibrium water flow in soil macropores as well as vertical heterogeneity in their hydraulic properties, while the drainage module also assumes horizontal saturated flow (Dupuit-Forcheimer assumptions) to a seepage surface above the drain (exactly as figure 11 shows).*

- Reply: The referee is right and we propose the following correction to complete the discussion dealing with performances of subsurface drainage models on clayey soils, after the line 507:

"Some models show their efficiency to simulate drainage discharge on such soils, as the MACRO model (Köhne et al., 2009), originally designed to simulate flow processes in structured soils like heavy clayey soils (see Fig. 12). However, these models are based on a more physically oriented approach than SIDRA-RU. Building on this type of concept would probably improve the latter simulating drainage discharges on clayey soils but would call into question its generalist nature."

Technical corrections:

The referee identified various technical corrections listed line by line that will be carried out as requested.

Dealing with the KGE' criterion, we cannot link the zero value to any particular interpretation unlike the NSE criterion (see Knoben et al. (2019): "there is no specific meaning attached to KGE'=0"). However, if we consider the mean flow benchmark as reference (corresponding to NSE = 0), the model must show a KGE' value above -0.41 to prove a better simulation than the mean flow (Knoben et al., 2019). We propose to include the following sentence into the text after the line 242 after "KGE' values range from $-\infty$ to 1":

"The model performance is better as KGE' increases towards 1. If the reader intends to use the mean flow benchmark as reference (corresponding to NSE = 0) to assess KGE', the target value is KGE' = -0.41 (Knoben et al., 2019)."

Regarding the English language issues, the article has already been corrected by a native speaker, as mentioned on the certificate attached to this document. However, the article will be reviewed a second time before the final submission to carry out the language issues highlighted by the referee.

References:

Bouarfa, S. and Zimmer, D.: Watertable shapes and drainflow rates calculation by Boussinesq's equation, 7th international drainage symposium in the 21st century : food production and the environment, Orlando, USA, 8-10 March 1998, 135, 1998.

Branger, F., Tournebize, J., Carluer, N., Kao, C., Braud, I., and Vauclin, M.: A simplified modelling approach for pesticide transport in a tile-drained field: The PESTDRAIN model, Agricultural Water Management, 96, 415–428, https://doi.org/10.1016/j.agwat.2008.09.005, 2009.

Henine, H., Jeantet, A., Chaumont, C., Chelil, S., Lauvernet, C., and Tournebize, J.: Coupling of a subsurface drainage model with a soil reservoir model to simulate drainage discharge and drain flow start, In publication.

Jarvis, N. J. and Leeds-Harrison, P. B.: Modelling water movement in drained clay soil. I. Description of the model, sample output and sensitivity analysis, Journal of Soil Science, 38, 487–498, https://doi.org/10.1111/j.1365-2389.1987.tb02284.x, 1987.

Jury, W. A. and Roth, K.: Transfer functions and solute movement through soil: theory and applications., in: Journal of plant nutrition and soil science, vol. 155, Issue 2, Birkhäuser Verlag AG, 77–166, 1992.

Knoben, W. J. M., Freer, J. E., and Woods, R. A.: Technical note: Inherent benchmark or not? Comparing Nash–Sutcliffe and Kling–Gupta efficiency scores, 23, 4323–4331, https://doi.org/10/ghvjxf, 2019.

Lagacherie, P. and Favrot, J. C.: Synthèse générale sur les études de secteurs de référence drainage, INRA, 1987.

Lesaffre, B.: France, in: Design practices covered drains in an agricultural land drainage system, Comité National Français de la CIID, 1987.

Magesan, G. N., Scotter, D. R., and White, R. E.: A transfer function approach to modeling the leaching of solutes to subsurface drains .I. Nonreactive solutes, Soil Res., 32, 69–83, https://doi.org/10/fwf5vw, 1994.

Robinson, M., Mulqueen, J., and Burke, W.: On flows from a clay soil—Seasonal changes and the effect of mole drainage, Journal of Hydrology, 91, 339–350, https://doi.org/10.1016/0022-1694(87)90210-1, 1987.

Snow, V. O., Houlbrooke, D. J., and Huth, N. I.: Predicting soil water, tile drainage, and runoff in a mole-tile drained soil, 50, 13–24, https://doi.org/10.1080/00288230709510278, 2007.

Tournebize, J., Chaumont, C., Marcon, A., Molina, S., and Berthault, D.: Guide technique à l'implantation des zones tampons humides artificielles (ZTHA) pour réduire les transferts de nitrates et de pesticides dans les eaux de drainage. Version 3, 2015.

Tuohy, P., Humphreys, J., Holden, N. M., and Fenton, O.: Runoff and subsurface drain response from mole and gravel mole drainage across episodic rainfall events, Agricultural Water Management, 169, 129–139, https://doi.org/10.1016/j.agwat.2016.02.020, 2016.

Robert Sachs
Translator / Copy-editor

March 3, 2021

To whom it may concern,

This certificate is intended to inform you that I, Robert Sachs, a professional French-English translator, proofreader and copy-editor with over 20 years experience working in close collaboration with the French research community, have reviewed and modified this article ("Robustness of a parsimonious subsurface drainage model at the French national scale"), on behalf of its lead author Alexis Jeantet. My assigned role was to correct the spelling / grammar / syntax of the manuscript in addition to improving its readability. I was naturally not involved in any of the strategic decisions regarding the article's outline, contents or dissemination of results. The scope of my function was primarily at the word / sentence level, in ensuring a quality of expression that would not detract from the authors' emphasis.

Furthermore, it is the role of the copy-editor to tighten the language, clean up syntax and avoid overt grammatical mistakes, in rendering a copy as cogent and efficient as possible for the reader, but not to realign sentences in a manner that would potentially subvert the authors' intended communication.

I remain at your entire disposal for any subsequent exchange you feel could be fruitful in the given context: robert.sachs@wanadoo.fr

Many thanks for your attention and consideration.

Best regards,

Robert Sachs
French Business Registration (SIRET):
399 113 786 00012

---

## Author Comment (AC2)

**Reply to the Referee 2**

We would like to thank the second referee for the relevant comments made on the manuscript and we are glad that the referee finds that this work may show an interest for the hydrological community of subsurface drainage. We agree with most of the comments and we propose an answer.

1. Regarding the clarity of English, the article has already been reviewed to correct language issues by a native speaker, as mentioned on the certificate attached to this document. However, the article will be reviewed a second time before the final submission to carry out the language issues highlighted by the referee;

2. The referee also points out that the text is a bit too long and needs to be shortened for clarity. The authors will strongly consider this point as requested. Some sentences from the Results part will be deleted such as the one from "The initial results" to "three main soil textures" at the line 299;

3. Eventually, the referee mentions a lack of rigour in the distinction between the Results and Discussion sections of the article. The authors will carry out this point limiting the Results section to the observations made on the graphs and tables without interpreting them. This will allow the text to be reduced in line with comment 2. Furthermore, some sentences from the Results part will be transferred to the Discussion part such as the one from "This result seems to be normal" to "the KGE' criterion" at the line 356.

The various specific comments will be addressed during the revision of the manuscript.

Robert Sachs
Translator / Copy-editor

March 3, 2021

To whom it may concern,

This certificate is intended to inform you that I, Robert Sachs, a professional French-English translator, proofreader and copy-editor with over 20 years experience working in close collaboration with the French research community, have reviewed and modified this article ("Robustness of a parsimonious subsurface drainage model at the French national scale"), on behalf of its lead author Alexis Jeantet. My assigned role was to correct the spelling / grammar / syntax of the manuscript in addition to improving its readability. I was naturally not involved in any of the strategic decisions regarding the article's outline, contents or dissemination of results. The scope of my function was primarily at the word / sentence level, in ensuring a quality of expression that would not detract from the authors' emphasis.

Furthermore, it is the role of the copy-editor to tighten the language, clean up syntax and avoid overt grammatical mistakes, in rendering a copy as cogent and efficient as possible for the reader, but not to realign sentences in a manner that would potentially subvert the authors' intended communication.

I remain at your entire disposal for any subsequent exchange you feel could be fruitful in the given context: robert.sachs@wanadoo.fr

Many thanks for your attention and consideration.

Best regards,

R. Sachs

Robert Sachs
French Business Registration (SIRET):
399 113 786 00012

---

## Author Response (AR1)

Response to the editor after its decision on July 03rd, 2021.

Robustness of a parsimonious subsurface drainage model at the French national scale

(hess-2021-168)

By A. Jeantet et al.,

Dear Dr. Stamm,

Thank you for your relevant comments on the manuscript after the discussion with the reviewers. We are glad that most of the proposed corrections to the reviewers' comments properly address the raised issues. They have integrated in the revised manuscript and highlighted by a specific comment mentioning the corresponding reviewer. The comments from N.JARVIS were detailed with the notation SC for "Specific Comment" and TC for "Technical Comment". A point-by-point answer is proposed for the remaining comments as well as a suggestion for correction accordingly hereafter:

Comment 2 from the reviewer 1:

The argument has been included as requested at the line 185 of the revised version:

"A sensitivity analysis on the SIDRA-RU model has revealed that $\alpha$ is not sensitive to the KGE' criterion (Henine et al., in review), used in this study as OF (see section 2.4.1), and moreover can be set at 1/3. Hence, to limit uncertainties relative to the calibration process for a non-sensitive parameter, this approach has been conserved herein."

Comment 6 from the reviewer 1:

A paragraph has been included in the Discussion part describing the potential application of the SIDRA-RU model for pollutant leaching and the required improvements or couplings with other modules at the line 601 of the revised version:

"In the perspective of long-term management on drained plots, predicting flows in order to better monitor the use of agricultural pollutants is a major concern, pollutant transfers occurring with drainage flow (Kladivko et al., 2001; Trajanov et al., 2018). Thus, a good model can be used as a decision-making tool, for example to restrict pollutants' application during flow period for the case of pesticides (Lewan et al., 2009; Zajíček et al., 2018; Kobierska et al., 2020). In this context, using SIDRA-RU may be quite relevant. However, the current form of the RU module is not optimal to accurately represent the fate of pollutant in soil profile, being too simple to precisely represent the behaviour of the water table inside the unsaturated zone. To overcome this problem, this model type is generally coupled with pedotransfer functions (Jury and Roth, 1992; Magesan et al., 1994) to transfer water and pollutant stock from the unsaturated zone to the saturated zone. Within this framework, the perspective of the PESTDRAIN module (Branger et al., 2009), coupled with the SIDRA-RU model, should allow simulating pesticide leaching by including two reservoirs: fast reservoir to mimic preferential flow above the drain area and slow reservoir through the matricial compartment. Based on a similar approach, combining SIDRA-RU with a nitrate leaching module might also be useful in order to correctly assess water pollution on French drained plots."

Comments for both reviewers regarding the linguistic issues:

The native speaker Robert Sachs has corrected the revised version of the manuscript, as attested by the certificate attached to this document. A final reading has been made on the manuscript to handle terms specific to the scientific field.

We hope this version fulfils your request and provides a fair correction for the raised issues.

Best regards,

Alexis Jeantet

Robert Sachs
Translator / Copy-editor

July 29, 2021

To whom it may concern,

This certificate is intended to inform you that I, Robert Sachs, a professional French-English translator, proofreader and copy-editor with over 20 years experience working in close collaboration with the French research community, have reviewed and modified this article ("Robustness of a parsimonious subsurface drainage model at the French national scale"), on behalf of its lead author Alexis Jeantet. My assigned role was to correct the spelling / grammar / syntax of the manuscript in addition to improving its readability. I was naturally not involved in any of the strategic decisions regarding the article's outline, subject matter or dissemination of results. The scope of my function was primarily at the word / sentence level, in ensuring a quality of expression that would not detract from the authors' emphasis.

As a resubmission to your journal, this article has already been reviewed by myself earlier in the year and, in the meantime, I have worked with Mr. Jeantet on its revisions in order to satisfy your quality specifications, in terms of both content and presentation.

I remain at your entire disposal for any subsequent exchange you feel could be fruitful for this particular submission: robert.sachs@wanadoo.fr

Many thanks for your attention and consideration.

Best regards,

Robert Sachs
French Business Registration (SIRET):
399 113 786 00012

---

## Editor Decision (ED1)

HESS 2021 168

Decision

Dear Dr. A. Jeantet

Thank you for submitted the revised version of your manuscript entitled
*"Robustness of a parsimonious subsurface drainage model at the French national
scale"*.

You have addressed many of the issues raised by the reviewers. The manuscript
seems considerably improved. Nevertheless, it was often not easy to identify
the actual changes in the text because no track-change-mode document was
provided. Please do so in the next round. In addition, there are still aspects
that require substantial improvements. Some of them got only evident with the
revised version. I list these issues below.

**Language:** Despite having the text read by a native speaker (according to your
letter), there are still wording issues throughout the manuscript. I list
some examples below to indicate the kind of problems that exist. Please
pay due attention to this aspect:

- L. 29: What the meaning of *exacerbating any infiltration concern*?

- L. 31 - 33: The expression *stone aggregates* sounds strange. Actually,
  the sentence is not properly citing the literature. Actually the word-
  ing is literally copied from the reference [1] but it refers NOT to mole
  drains but to ditches separating the experimental plots. The article
  itself deals with mole drains and gravel mole drains. Accordingly, the
  sentence on L. 32 is factually wrong. Please correct.

- L. 52: *... thus complicating their parameterization on a large database..*
  Weird sentence, which is not clear. Please rephrase.

- L. 275 - 275: This information is repeated for the third time. Please
  skip and avoid unnecessary repetitions throughout the manuscript.

- L. 301 - 309: This information has already be provided in the previous
  paragraph. Skip.

- L. 353 - 361: Not the $Q_{05}$ are close to zero, but the respective biases!
  Please adapt the wording for all quantiles!

- L. 430: This sentence is not correct. I assume it should be: *Not all
  regions .... are well represented.*

- L. 452: *respected*: this is not the adequate expression here.

- L. 459: *respects*: Rephrase: simulates well the temporal ...

- L. 467: Reword. There are not just two extreme discharges.

- L. 490: Why should *model calibration (be) only relevant if the calibrated K and μ are probable according to the case study soil type*? The word *relevant* seems inappropriate here.

- L. 516: *relevant*: Not an appropriate term: it is also relevant if the performance is poor at a given site. Relevant has the meaning of be connected with the matter at hand (see e.g., `https://www.wordreference.com/definition/relevant`).

- L. 547: What is the robustness approach? Strange wording, please rephrase.

- L. 551: What is a conventional period?

- L. 566: *Adopting assumptions on model performance would then offer a relevant alternative*: What does this mean?

- L. 568: Reword: ... *that also limit the study.* (I assume that the assumptions were not made in order to serve as limitations!).

- L. 571: Re-word: *in every other context.* Your current version tells that nowhere else the assumption were true while you want to say that not everywhere it was true.

- L. 612: *matricial compartment*: replace by soil matrix.

**Calibration:**
- L. 65 - 72: What are the arguments for selecting KGE in your case?

- L. 227 - 231: Where can one see the priors? Provide these data.

- L. 441 - 442: Is it actually necessary to use these priors? Wouldn't the calibration process find the appropriate values anyway? Please test this aspect if haven't done already.

- L. 578: What about the priors?

- L. 586 - 587: Where can one see these data?

- L. 587 - 592: Are the calibrated parameters indeed constraint by the priors?

**Further comments:**
- L. 43: This re-design of drainage system is not further elaborated on in the manuscript. Can you comment on that aspect in the discussion when talking about the benefits of the model?

- L. 53: What is *semi-conceptual*? Please be consistent in your terminology. Often you describe the model as conceptual (e.g., L. 258; which seems clear to me).

- L. 54: You should make it explicitly clear that the model has six parameters. In your application described here, you simply kept two parameters fixed at values obtained from prior knowledge. This is important for applications elsewhere (see L. 173 - 188, also L. 439).

- L. 90: Provide examples of what you do not consider. Discuss what the implications could be.

- L. 95 - 96: But this would also be possible with other classifications?
- L. 108 - 109: *where drained soils are predominantly silty-clayey and composed of fine sediment with heavy clay*: this is confusing. Is the texture silty-clayey or heavy clay? It cannot be both at the same time, can it?
- L: 119: *drainage modalities*: what does this mean?
- L. 204 - 208: In your response you mention that the model could also deal with leaky subsoils. Please be specific about that and briefly describe how the model could represent this situation.
- L. 218 - 219: *In this context, the model calibration allows estimating parameters based on a comparison between observations and model simulations.* This is trivial and can be skipped.
- L. 322: Is this a general finding or site-specific? If the peaks are underestimated but the overall flux is ok, it implies that the recession curve overestimates water flow. Is this correct and a general observation or is it site-specific?
- Fig. 8. I assume you have two data points per site from the split test. Please indicate (by specific symbols) which data points belong to the same site.
- L. 437: Why is a simple model better for generalizing? A physics-based model should be better suited because it can accommodate more different situations while a conceptual model imposes more implicit model constraints. This view is supported by your description of the conceptual limitations of SIDRA-RU. On L. 521 - 522 you explicitly state that the model was primarily designed for silty soils and the results clearly demonstrate the limitations that go with that. You illustrate this also very clearly in Fig. 12. This all seems to contradict you claim that a simpler conceptual model is better suited for general applications.
- L. 451: *more local studies*: provide references.
- L. 471 - 472: *This rationale might partially explain the observed delay.*: You can simply test this explanation instead of being speculative.
- L. 500: What is the metric for consistency? There are quite some deviations.
- Fig. 11: If I understand correctly, you have a single parameter value for each site from the reference data base and the calibration (actually, from the calibration you should have two from the split test). Accordingly, it would be more informative to compare the site-specific values.
- L. 519: N. Jarvis commented on the good performance of MACRO on such soils. This has to be mentioned and referenced to avoid a wrong impression.

- L. 544: Why should that be the case? As you describe, SIDRA-RU is not that general because it performs not very well on heavy clay soils (due to conceptual constraints, see also L 521 - 522, Fig. 12). Why should MACRO perform poorly on non-heavy clay soils? The preferential flow part of MACRO is not a static feature but depends on site-specific parameters.

- L. 550 - 551: Why should the calibration period a dry period? Where can one see this?

- L. 552 - 554: This is a hypothesis that can be tested on your data. Please do so.

- L. 558 - 561: Above you argue that crops are not relevant. This is now confusing.

- L. 572 - 573: Is this already implemented or does it need a change in the model code?

- L. 604 - 605: *Thus, a good model can be used as a decision-making tool, for example to restrict pollutants' application during flow period for the case of pesticides*: It is a typical claim to state that a model is required to support decision making. But is it actually true? Practitioners generally know very well flow periods and if they were to restrict e.g., pesticide or fertilizer applications during such periods, models weren't necessary, I'd argue. In such a situation it most probably not the lack of knowledge regarding the flow regime that limits such restrictions but the crop-specific timing and needs for nutrients or crop protection. Please be more precise in describing what can be achieved for practice for which there is hardly an alternative for models.

- L. 608 - 609: How do transfer function solve this problem? You need to come up with an adequate transfer function model. A key aspect is that water flow is not identical to solute transport. This implies that conceptually, one has to add components such as to account for the transport aspect. Important in this context: fast transport is of high relevance for sorbing (and degrading) compounds such as P or pesticides. Hence, even if surface runoff may be irrelevant for simulating the water fluxes and the water balance, it may be essential to account for runoff (and preferential flow to tile drains) in the model concept. Often this implies that one has to introduce a shallow topsoil layer to account for the crucial processes (sorption, degradation, mobilisation) controlling the fate of agrochemicals. Refer to relevant conceptual models in the literature to provide some more depth to the discussion.

- L. 618: The term *exhaustive* seems inadequate given the model limitations on e.g., clay soil that are mentioned above.

Please address these issues in an adequate form.

Sincerely

Dr. Christian Stamm Editor HESS

**References**

[1] Tuohy, P., Humphreys, J., Holden, N. M., & Fenton, O. (2015). Mole drain performance in a clay loam soil in Ireland. Acta Agriculturae Scandinavica, Section B — Soil & Plant Science, 65(sup1), 2-13. doi:10.1080/09064710.2014.970664

---

## Author Response (AR2)

Response to the editor after its decision on August 17th, 2021.

Robustness of a parsimonious subsurface drainage model at the French national scale

(hess-2021-168)

By A. Jeantet et al.,

Dear Dr. Stamm,

Thank you for your valuable comments improving clearly the manuscript after receiving the revised version. They have been integrated in the revised manuscript and highlighted by a specific comment mentioning the corresponding order of appearance. The mention "Language" designates the comments from the Language part, "Calibration" for those from the Calibration part and "FC" for those from the "Further comments" part. The lines mentioned-below refer to the marked version:

**"Language":**

All the language issues have been corrected as requested.

1. *"L. 29: What the meaning of exacerbating any infiltration concern?"* Reply: The expression has been replaced at the line 29 by:

"reducing the deep infiltration"

 "L. 31 - 33: The expression stone aggregates sounds strange. Actually, the sentence is not properly citing the literature. Actually the wording is literally copied from the reference [1] but it refers NOT to mole drains but to ditches separating the experimental plots. The article itself deals with mole drains and gravel mole drains. Accordingly, the sentence on L. 32 is factually wrong. Please correct." Reply: To avoid useless descriptions, the sentence at lines 33-34 has been removed and replaced by:

"such as subsurface drainage and open ditch."

"L. 52: ... thus complicating their parameterization on a large database. Weird sentence, which is not clear. Please rephrase."
 Benly: The line 54 has been complete by:

Reply: The line 54 has been complete by:

"with large number of parameters. Their calibration on several study sites becomes difficult and time consuming (Beven, 1989)"

- "L. 275 275: This information is repeated for the third time. Please skip and avoid unnecessary repetitions throughout the manuscript." Reply: The line 275 has been removed.
- "L. 301 309: This information has already be provided in the previous paragraph. Skip." Reply: The paragraph from the lines 313 to 321 has been removed and merged with the previous paragraph, from the lines 302 to 312:

"Table 1 lists the performance over the entire calibration period obtained from all 22 sites and Table 2 classifies the model performance from each soil texture according to the score ranges. Performance varies across the three soil textures, with both unsatisfactory KGE' values, e.g. for the *Courcival\_P3* site, and some "very good" KGE' values, e.g. *Parisot*. For 21 of the 22 referenced plots, the calibration KGE' lies above 0.5, thus revealing at least "acceptable" KGE' values. The silty plots show values ranging from 0.54 to 0.83, including the best model performances, such as *La\_Jaillière\_P4* plot, with a KGE' of 0.83. They compile three "acceptable" scores, reaching "good" for six of them and "very good" for another six. The silty-clayey plots exhibit relatively homogenous KGE' values, ranging from 0.54 to 0.76. As regards

the clayey plots, KGE' values display a wider range than on the silty-clayey plots, i.e. from 0.44 at the *Courcival\_P3* plot to 0.76 at *Saint\_Laurent\_P2* but the model performance remains at least "acceptable" on most of them (including one "very good"). *Courcival\_P3* is the only one indicating an "unsatisfactory" KGE' value"

6. "L. 353 - 361: Not the Q05 are close to zero, but the respective biases! Please adapt the wording for all quantiles!"

Reply: The paragraph from the lines 366 to 377 has been reworded:

"Regarding the Q05 quantiles (Fig. 7a), results show that for the three textures, bias between simulated and observed Q05 ranges from -0.020 to 0.030 mm/d, with some extreme points (mainly on the silty texture). The medians of biases all lie close to zero as well (from -0.002 to 0.002 mm/d), thus revealing that the model correctly predicts low flows. Regarding the Q95 quantiles (Fig. 7b), the median values from boxplots are once again close to zero (from -0.247 to -0.040 mm/d); however, the ranges of the Q95 biases lie above those of the Q05 quantiles. On silty soils, the boxplot limits of Q95 biases range from -1 mm/d to + 0.5 mm/d, and the whiskers range from -3 mm to +3 mm. Similarly, for silty-clayey soils, the Q95 biases vary from -3 mm to -2 mm; the discrepancies are larger on clayey soils, where the Q95 biases varies from -4 mm to +4 mm. Figure 7c shows that the boxplot medians for the Qmean biases also lie close to zero (from 0.007 to 0.057 mm/d). The Qmean biases range from -0.5 mm to +0.5 mm for silty soils, from -0.3 mm to +0.6 mm for silty-clayey soils, and from -0.8 mm to +0.9 mm for clayey soils. SIDRA-RU performs at a level of good agreement with respect to the average drainage discharges. The deviation on Qmean biases is higher on clayey soils, thus reflecting the greater difficulties of the SIDRA-RU model in simulating Qmean on this texture."

7. *"L. 430: This sentence is not correct. I assume it should be: Not all regions .... are well represented."* Reply: The sentence at lines 444-445 has been replaced by :

"Not all regions with a high drainage rate are well represented."

8. *"L. 452: respected: this is not the adequate expression here."* Reply: "respected" has been replaced by:

"reproduced with a good agreement"

9. *"L. 459: respects: Rephrase: simulates well the temporal ..."* Reply: The sentence at lines 476-477 bas been reworded:

"Also, the SIDRA-RU model allows simulating the temporal variation of drainage discharge for both dry and wet periods;"

- 10. *"L. 467: Reword. There are not just two extreme discharges."* Reply: Line 484, the expression "at the two" has been replaced by the word "on".
- 11. "L. 490: Why should model calibration (be) only relevant if the calibrated K and μ are probable according to the case study soil type? The word relevant seems inappropriate here."
  Reply: the editor is right; the word "relevant" has been replaced by "reliable" at the line 508.
- 12. "L. 516: relevant: Not an appropriate term: it is also relevant if the performance is poor at a given site. Relevant has the meaning of be connected with the matter at hand (see e.g., https://www.wordreference.com/definition/relevant)." Reply: The editor is right; the word "relevant" has been replaced by "good".
- 13. *"L. 547: What is the robustness approach? Strange wording, please rephrase."* Reply: As requested, the lines 567-569 have been reworded as follow:

"Another limitation of this model is the RU parameters are slightly less temporally robust than the SIDRA parameters, showing fewer stable values between calibration sub-periods than the SIDRA parameters."

14. "L. 551: What is a conventional period?"

Reply: Here, a conventional period refers to a common hydrological period without extreme events. To improve the clarity of the discussion, the sentence at lines 571-573 has been reworded:

"Compared to a common hydrological period without too much extreme discharge events, a dry calibration period conducts to increase  $S_{inter}$  values"

15. *"L. 566: Adopting assumptions on model performance would then offer a relevant alternative: What does this mean?"*

Reply: The sentence at lines 590-591 has been reworded:

"Making assumptions, e.g. grouping soils by categories, to address this weaknesses would then offer a good alternative"

16. "L. 568: Reword... : that also limit the study. (I assume that the assumptions were not made in order to serve as limitations!)."

Reply: The sentence from the line 593 to 596 has been reworded:

"Moreover, the model is based on some rather important assumptions, such as neglecting both the surface runoff and recharge to groundwater, or the ones made on the parameters a and  $\alpha$ , which limit its field of application. Indeed, "

17. "L. 571: Re-word: in every other context. Your current version tells that nowhere else the assumption were true while you want to say that not everywhere it was true."Reply: As requested, the sentences at lines 597-598 have been reworded:

"up to now there is no evidence of their relevance on other sites. This limits the use of SIDRA-RU to the specific conditions outlined herein"

 "L. 612: matricial compartment: replace by soil matrix." Reply: As requested, the term "matricial compartment" has been replaced by "soil matrix" at the line 641.

**"Calibration":**

1. *"L. 65 - 72: What are the arguments for selecting KGE in your case?"* Reply: The argument justifying to KGE' is detailed at lines 245-246:

"We have thus introduced the KGE' criterion (Kling et al., 2012), an evolution of KGE that is more relevant than NSE in reproducing internal flow rate variability (Santos et al., 2018)"

To avoid repetitions, we did not integrate this argument in the introduction part.

 "L. 227 - 231: Where can one see the priors? Provide these data." Reply: The editor is right; the characteristics of distribution are missing. A table grouping the mean and standard deviation bas been integrated in the Appendix part and the following sentence has been integrated at lines 237-240:

"The mean and standard deviation of each parameter are available in Appendix A. The ones for K and  $\mu$  were extracted per soil texture from the aforementioned reference drainage areas. The ones for Sinter and Smax were numerically fixed after many calibration tests."

- "L. 441 442: Is it actually necessary to use these priors? Wouldn't the calibration process find the appropriate values anyway? Please test this aspect if haven't done already."
   Reply: Using the priors is necessary, the calibration algorithm searches in each parameter space to find the best combination. This requires a prior knowledge of the corresponding distributions. The calibration process cannot be performed without this information.
- 4. "Line 578: What about the priors?"

Reply: The calibration algorithm is designed to be as automatic as possible (Mathevet, 2005). However, it requires the distributions of each parameter, which might introduce biases. This aspect is highlighted at lines 605-606: "external decisions influencing results are still necessary" and developed at lines 614-616 from "These measurements are subjected to uncertainty" to "that bias the mean and standard deviation".

- "L. 586 587: Where can one see these data" Reply: These data are extracted from 96 reports on reference drainage tests, provided by INRAE on request. Vincent (1989) provides a list of technical datasheets of all these experiments and is available as requested or in university libraries.
- 6. *"L.* 587 592: Are the calibrated parameters indeed constraint by the priors" Reply: The editor is right; the calibrated parameters are constraint by the distributions used in the calibration algorithm. To detail this aspect, we propose to include the following sentence at the line 617 after "the calibrated parameters" and before "might be biased":

", constraint by the distributions a priori defined,".

**Further comments:**

- "L. 43: This re-design of drainage system is not further elaborated on in the manuscript. Can you comment on that aspect in the discussion when talking about the benefits of the model?" Reply: The editor is right; this aspect is not further specifically mentioned in the manuscript after this sentence. The authors consider that it deals with the evolution of subsurface drainage under climate change, which is not the topic of the manuscript. Another paper dealing with this aspect is currently under correction and will specifically analyse the becoming of French subsurface drainage under climate change.
- 2. "L. 53: What is semi-conceptual? Please be consistent in your terminology. Often you describe the model as conceptual (e.g., L. 258; which seems clear to me)."

Reply: The "semi-conceptual" term refers to the fact that the SIDRA-RU model is composed by a physically-based module (the historical SIDRA module) coupled with a conceptual module (the RU module) (Beskow et al., 2011). Consequently, the SIDRA parameters K and  $\mu$  are physically-based instead of the RU parameters Sinter and Smax are conceptual. To improve the clarity of this aspect, the following correction have been applied:

• to rewrite the sentence at lines 56-58 in the Introduction part:

"The model is semi-conceptual (Beskow et al., 2011), being composed of one physically-based part (the SIDRA module) coupled with a conceptual part (the RU module) and parsimonious".

• to remote the sentence "such as SIDRA-RU" at the line 267.

3. "L. 54: You should make it explicitly clear that the model has six parameters. In your application described here, you simply kept two parameters fixed at values obtained from prior knowledge. This is important for applications elsewhere (see L. 173 - 188, also L. 439)."

Reply: The editor is right; the manuscript has been corrected as follow:

• At the line 58, the expression "4 parameters" has been replaced by "six parameters";

- The sentence at lines 191-192 has been rewritten as follow: "The parameter α defines the proportion of Pnet(t) being converted to recharge R(t), while the remainder updates the water level, i.e. Eq. (3)";
- The sentence at lines 215-216 has been rewritten as follow: "due to the aforementioned assumptions dealing with the parameters a and a, a calibration process is only necessary for four parameters";
- The sentence at lines 454-455 has been rewritten as follow: ", initially requiring the calibration of six parameters, the assumptions made in this study allow to reduce this number to four parameters";
- The following sentence at the line 595 has been included: ", or the ones made on the parameters  $\alpha$  and  $\alpha$ ,".
- 4. *"L. 90: Provide examples of what you do not consider. Discuss what the implications could be."* Reply: The editor is right, some soil characteristics are not considered. To improve this aspect, the following paragraph from the line 92 to the line 101 has been rewritten as follow:

"A multitude of materials constitute French soils, as defined by their geological context, textural evolution and regional climate. All of the above characteristics serve to determine the uniqueness of a soil. Making generalizations about soil diversity becomes then a necessary step. Indeed, grouping them by soil category facilitates their modeling. Several official classifications serve to group soil types (FAO, 1988; Krogh and Greve, 1999; Driessen et al., 2000). In this study, we are proposing to classify them by texture, thus making it possible to sort the database into three categories (see Fig. 1 & Table 1). Let's note that here we do not consider the geological context or the regional climate to classify soils."

Section 4.7 already discusses of the consequences of such oversights, highlighting that grouping soils only using their soil texture introduces biases. To avoid repetitions, no other sentence has been integrated dealing with this aspect.

5. "*L.* 95 - 96: But this would also be possible with other classifications?" Reply: The editor is right; theoretically, the classification can be made using other characteristics as discussed in the previous comment. However, this would be in practice difficult because the classification would be too restrictive, e.g. to obtain groups with one soil which is not relevant. Regarding the number of available sites in the used database, this risk is non-negligible.

6. "L. 108 - 109: where drained soils are predominantly silty-clayey and composed of fine sediment with heavy clay: this is confusing. Is the texture silty-clayey or heavy clay? It cannot be both at the same time, can it?"

Reply: The editor is right; in our classification, a plot cannot be simultaneously silty-clayey and composed of heavy clayey. However, two plots of the same study site can be defined by two different soil textures as *La Bouzule*. To avoid confusions, the sentence at lines 113-114 has been corrected as follow:

"where drained soils are either predominantly silty-clayey or heavy clayey soils."

7. "L: 119: drainage modalities: what does this mean?" Reply: The term "Drainage modalities" refers here to the technical aspects of drainage networks, such as depth, spacing or the material used for the pipes. The reference drainage tests were performed to define what practice best suits to the conditions of a given site. We propose to include the following sentence after "drainage modalities" at lines124-125:

", i.e. what depth, space or pipes best fit to the field conditions".

8. "L. 204 - 208: In your response you mention that the model could also deal with leaky subsoils. Please be specific about that and briefly describe how the model could represent this situation."

Reply: The editor is right; the model could deal with leaky soils integrating the depth infiltration with Hooghoudt's equation, according to the principle of equivalent depth (Zimmer, 1992), introducing the term "Ds", designating the deep seepage rate, in the equation to reduce the recharge term to the drains. This aspect is already mentioned at lines 599-600, and we propose to replace the reference by (Zimmer, 1992) and complete the sentence by:

"A term " $D_s$ ", designating the deep seepage rate, is introduced in the Boussinesq's equation to reduce the recharge rate to the drains."

- 9. "L. 218 219: In this context, the model calibration allows estimating parameters based on a comparison between observations and model simulations. This is trivial and can be skipped." Reply: The editor is right; the sentence has been deleted.
- 10. "L. 322: Is this a general finding or site-specific? If the peaks are underestimated but the overall flux is ok, it implies that the recession curve overestimates water flow. Is this correct and a general observation or is it site-specific"

Reply: The editor is right; it is a general trend. The model quite underestimates peak flows and simulate longer recession periods. However, the intensity of this phenomenon varies from one site to another and we cannot relate it to any soil characteristic. The phenomenon seems to depend on the calibration quality. Let's not that the model better simulates peak flows at the hourly time step introducing a new term in the Boussinesq's equation (Bouarfa and Zimmer, 1998).

11. "Fig. 8. I assume you have two data points per site from the split test. Please indicate (by specific symbols) which data points belong to the same site."Reply: The editor is right; there are two points per site from the split sample test. Initially, the purpose was to illustrate each site by a single symbol. However, the split sample test is performed on 9 sites, which imposes 9 different symbols, not facilitating the understanding of the graph. Eventually, only the

which imposes 9 different symbols, not facilitating the understanding of the graph. Eventually, only the soil texture has been represented. If the reader wants to compare the KGE' values specific to a single site, he can refer to Table 1 which provides the KGE' values in calibration and in evaluation from the split-sample test.

12. "L. 437: Why is a simple model better for generalizing? A physics-based model should be better suited because it can accommodate more different situations while a conceptual model imposes more implicit model constraints. This view is supported by your description of the conceptual limitations of SIDRA-RU. On L. 521 - 522 you explicitly state that the model was primarily designed for silty soils and the results clearly demonstrate the limitations that go with that. You illustrate this also very clearly in Fig. 12. This all seems to contradict you claim that a simpler conceptual model is better suited for general applications."

Reply: The editor is right; a physically-based model is theoretically better suited to represent a large soil diversity. However, this kind of model is generally composed of many parameters representing the complexity of a study site such as current crop, root depth, saturated water content (i.e. unsaturated one), hydraulic parameters or water holding capacity. Performing such a model on a large database requires providing all these characteristics for each site, which is in practice very difficult, because the measurement technics are globally expensive. The calibration might be useful but it is time consuming due to their large number of parameters of such models. Indeed, a simple model offers significant advantages, as it requires few information and faster calibration, thus becoming in practice suitable for generalizing. Let's note that, in line with what the editor remarks, a simple model is limited by the assumptions made on its design.

13. "L. 451: more local studies: provide references."

Reply: As requested by the editor, the following references are provided at the line 467: (Gowda et al., 2012; Skaggs et al., 2012; Muma et al., 2017; Revuelta-Acosta et al., 2021).

14. "L. 471 - 472: This rationale might partially explain the observed delay.: You can simply test this explanation instead of being speculative."

Reply: The editor is right; it would be better to accurately highlight the origin of bias at the start of the drainage season performing tests. However, this starts is controlled by many factors depending on the climate conditions of the past year, the actual ones and the soil conditions. All of these factors influence the Sinter parameter, which is calibrated on the entire calibration period. Consequently, to prove this phenomenon is quite difficult. As proposed in Section 4.6 at lines 578-580, to make Sinter dependent on

these annual conditions, we can adjust it on every year according to the aforementioned factors. To avoid any confusion, we propose to remove the sentence at the line 488-489: "This rationale might partially explain the observed delay".

15. "L. 500: What is the metric for consistency? There are quite some deviations."

Reply: The editor is right; there are some quite deviations. However, the distributions are in the same order of magnitude. Regarding the hydraulic conductivity K, except for three calibrated sites (20% of the calibrated sites) showing high values compared to the reference sites, the distributions on the remaining 12 sites are similar. The deviation in mean value does not exceed 0.05 m/d. In this regard, the histograms from the calibrated K are congruent with the ones from reference drainage tests. We propose to correct the sentence from the lines 516-517 by:

"However, over the remaining range of values, the histogram from the calibrated K are congruent with the one from the reference drainage tests".

- 16. "Fig. 11: If I understand correctly, you have a single parameter value for each site from the reference data base and the calibration (actually, from the calibration you should have two from the split test). Accordingly, it would be more informative to compare the site-specific values."
  Reply: The editor is right; we have one value per reference site and one value per calibrated site and it would be better to directly compare the site-specific values. However, we do not have a sufficient number of measured values of K and μ from the calibrated sites to do so.
- 17. "L. 519: N. Jarvis commented on the good performance of MACRO on such soils. This has to be mentioned and referenced to avoid a wrong impression."
  Reply: The performance of the MACRO model on clayey soils has already been mentioned at lines 559-561 as requested by N. Jarvis (see comment 18 from the "Further Comments" part): "Among them, there is the MACRO model which showed its efficiency to simulate drainage discharge in Europe, including structured soils like heavy clayey soils (Köhne et al., 2009)".
- 18. "L. 544: Why should that be the case? As you describe, SIDRA-RU is not that general because it performs not very well on heavy clay soils (due to conceptual constraints, see also L 521 522, Fig. 12). Why should MACRO perform poorly on non-heavy clay soils? The preferential flow part of MACRO is not a static feature but depends on site-specific parameters."
  Barlyn The aditor is right to swild applying any constrainty of parameters.

Reply: The editor is right; to avoid confusions, we propose to correct the sentence at lines 559-561 by:

"Among them, there is the MACRO model which showed its efficiency to simulate drainage discharge in Europe, including structured soils like heavy clayey soils (Köhne et al., 2009)"

- 19. "L. 550 551: Why should the calibration period a dry period? Where can one see this?" Reply: Dry periods might appear due to climate conditions, observing a larger occurrence of dry events on a same period. In practice, the model might be calibrated on such a period. Figure 5 shows that at *Courcival\_P3*, the period 1991 to 1995, on which the model is calibrated for the split-sample test, is drier than the one from 1986 to 1990.
- 20. "L. 552 554: This is a hypothesis that can be tested on your data. Please do so."

Reply: The editor is right. Regarding *Courcival\_P3*, the split-sample test showed that Sinter is stronger on the second period from 1991 to 1995 (175 mm) than on the first period 1986 to 1990 (202 mm). As above-mentioned, the second period is drier. We propose the following correction:

- To remove "and Smax" at the line 575;
- To integrate this example at lines 575-576:

"Regarding *Courcival\_P3*, the split-sample test showed that Sinter decreased from 175 mm on 1986-1990 to 202 mm on 1990-1995, a dry period (see Fig. 5)."

- 21. "L. 558 561: Above you argue that crops are not relevant. This is now confusing." Reply: The editor is right; to avoid confusions, we propose to remove the entire paragraph from lines 582 to 585.
- 22. "L. 572 573: Is this already implemented or does it need a change in the model code?"

Reply: The current version of SIDRA-RU requires a change in the model code to integrate deep seepage. The latter is managed in an older version of the SIDRA module (see comment 8 in the "Further Comments" part).

23. "L. 604 - 605: Thus, a good model can be used as a decision-making tool, for example to restrict pollutants' application during flow period for the case of pesticides: It is a typical claim to state that a model is required to support decision making. But is it actually true? Practitioners generally know very well flow periods and if they were to restrict e.g., pesticide or fertilizer applications during such periods, models weren't necessary, I'd argue. In such a situation it most probably not the lack of knowledge regarding the flow regime that limits such restrictions but the crop-specific timing and needs for nutrients or crop protection. Please be more precise in describing what can be achieved for practice for which there is hardly an alternative for models."

Reply: The editor is right; a better knowledge dealing with the crop-specific timing and needs for nutrients or crop protection could be an efficient way to protect agricultural water against pollutants. However, to be able to anticipate the start of flow might also have a major contribution in such a context. Lewan et al. (2009) attested that this is the one of the most efficient strategies helping farmers to reduce pesticide transfer risk. Some solutions recommend to use restriction based on water content instead of restriction timing or ban (Brown and van Beinum, 2009; Lewan et al., 2009). We proposed to add the following sentence at lines 633-635:

"Indeed, using water content as an indicator to anticipate the start of drainage flow in order to reduce pesticide applications is a recommended strategy instead of restriction timing (Brown and van Beinum, 2009; Lewan et al., 2009)"

24. "L. 608 - 609: How do transfer function solve this problem? You need to come up with an adequate transfer function model. A key aspect is that water flow is not identical to solute transport. This implies that conceptually, one has to add components such as to account for the transport aspect. Important in this context: fast transport is of high relevance for sorbing (and degrading) compounds such as P or pesticides. Hence, even if surface runoff may be irrelevant for simulating the water fluxes and the water balance, it may be essential to account for runoff (and preferential flow to tile drains) in the model concept. Often this implies that one has to introduce a shallow top-soil layer to account for the crucial processes (sorption, degradation, mobilisation) controlling the fate of agrochemicals. Refer to relevant conceptual models in the literature to provide some more depth to the discussion."

Reply: The editor is right; we totally agree with this summary of solute transport in tile-drained soils. The transfer functions will not fully solve the problem, compared to the more physically-based models mentioned-above, but constitute a simple approach to be coupled with the SIDRA-RU model. In this framework, choosing an adequate transfer function is strongly required. Regarding the PESTDRAIN module (Branger et al., 2009), two reservoirs are integrated, one for fast transport and one for slow transport, and both of them use their own transfer functions, being exponential type. Furthermore, PESTDRAIN already integrates surface runoff, required step as highlighted by the editor. We believe that citing Magesan et al. (1994) and Branger et al. (2009) is sufficient.

25. "L. 618: The term exhaustive seems inadequate given the model limitations on e.g., clay soil that are mentioned above."Reply: The term "exhaustive" has been replaced by "large" at the line 648.

We hope this version fulfils your requests. We added a sentence in the Acknowledgments part to thank you and the reviews for your relevant and valuable comments, which strongly improved the quality of this paper.

Best regards,

Alexis Jeantet

**References:**

Beskow, S., Mello, C. R., Norton, L. D., and da Silva, A. M.: Performance of a distributed semi-conceptual hydrological model under tropical watershed conditions, CATENA, 86, 160–171, https://doi.org/10/dvz95p, 2011.

Bouarfa, S. and Zimmer, D.: Watertable shapes and drainflow rates calculation by Boussinesq's equation, 7th international drainage symposium in the 21st century : food production and the environment, Orlando, USA, 8-10 March 1998, 135, 1998.

Branger, F., Tournebize, J., Carluer, N., Kao, C., Braud, I., and Vauclin, M.: A simplified modelling approach for pesticide transport in a tile-drained field: The PESTDRAIN model, Agricultural Water Management, 96, 415–428, https://doi.org/10/bxbn6p, 2009.

Brown, C. D. and van Beinum, W.: Pesticide transport via sub-surface drains in Europe, Environmental Pollution, 157, 3314–3324, https://doi.org/10/fsjg3b, 2009.

Gowda, P. H., Mulla, D. J., Desmond, E. D., Ward, A. D., and Moriasi, D. N.: ADAPT: Model use, calibration, and validation, 55, 1345–1352, https://doi.org/10/f39v9d, 2012.

Jannot, Ph.: Drainage and crop production system on intensive dairy farms in Western France, Agricultural Water Management, 14, 61–68, https://doi.org/10/d3z76b, 1988.

Köhne, J. M., Köhne, S., and Šimůnek, J.: A review of model applications for structured soils: a) Water flow and tracer transport, Journal of Contaminant Hydrology, 104, 4–35, https://doi.org/10/fbdvsh, 2009.

Lagacherie, P. and Favrot, J. C.: Synthèse générale sur les études de secteurs de référence drainage, INRA, 1987.

Lewan, E., Kreuger, J., and Jarvis, N.: Implications of precipitation patterns and antecedent soil water content for leaching of pesticides from arable land, Agricultural Water Management, 96, 1633–1640, https://doi.org/10/cx2jb9, 2009.

Magesan, G. N., Scotter, D. R., and White, R. E.: A transfer function approach to modeling the leaching of solutes to subsurface drains .I. Nonreactive solutes, Soil Res., 32, 69–83, https://doi.org/10/fwf5vw, 1994.

Mathevet, T.: Quels modèles pluie-débit globaux au pas de temps horaire? Développements empiriques et comparaison de modèles sur un large échantillon de bassins versants, Thèse de Doctorat, à l'Ecole Nationale du Génie Rural, des Eaux et Forêts, 2005.

Muma, M., Rousseau, A. N., and Gumiere, S. J.: Modeling of subsurface agricultural drainage using two hydrological models with different conceptual approaches as well as dimensions and spatial scales, 42, 38–53, https://doi.org/10/ghvj2j, 2017.

Revuelta-Acosta, J. D., Flanagan, D. C., Engel, B. A., and King, K. W.: Improvement of the Water Erosion Prediction Project (WEPP) model for quantifying field scale subsurface drainage discharge, 244, https://doi.org/10/ghksw4, 2021.

Skaggs, R. W., Youssef, M., and Chescheir, G. M.: DRAINMOD: model use, calibration, and validation, Transactions of the ASABE, 55, 1509–1522, https://doi.org/10/f39wqd, 2012.

Vincent, B.: Drainage. Secteurs de référence drainage. Recueil des expérimentations, CEMAGREF-DICOVA., CEMAGREF Editions, Antony, 186 pp., 1989.

Zimmer, D.: Effect of deep seepage on drainage functioning and design in shallow soils, 6th international drainage symposium "Drainage and water table control", Nashville, USA, 13-15 December 1992, 272, 1992.

---

## Editor Decision (ED2)

hess-2021-168-manuscript-version5

Decision

Dear Dr. A. Jeantet

Thank you for revising carefully your manuscript entitled *"Robustness of a parsimonious subsurface drainage model at the French national scale"*. You have addressed most of the points that I had listed before. Still, I think that seven points need some more clarification or improvement. I describe them below and refer to your enumeration in your response.

**Language:**

    # 5 (L. 310 in the track-change document): Replace *compile* by *include*.

    # 6: Consistently use $mmd^{-1}$ instead of $mm$ only. On L. 372, skip *from boxplots*: the median is not derived from these plots.

    # 14: Rephrase as ... *period increases $S_{inter}$ values.*

**Further comments:**

    # 10: Please mention this general trend in the text.

    # 12: Please specify your statement in the paper according to your response.

    # 16: I cannot follow this argument: for each site, the is a value for each parameter from the the reference data base and the respective calibrated value. These values can be compared. Please do so. You may add the results in the Appendix.

    # 19: Please provide the argument also in the paper.

Please address these issues.
Sincerely

Dr. Christian Stamm Editor HESS

---

## Author Response (AR3)

Response to the editor after its decision on September 10th, 2021.

Robustness of a parsimonious subsurface drainage model at the French national scale

(hess-2021-168)

By A. Jeantet et al.,

Dear Dr. Stamm,

Thank you for your comments improving the manuscript after receiving the revised version. As the previous version, they have been integrated in the revised manuscript and highlighted by a specific comment mentioning the corresponding order of appearance. The mention "Language" designates the comments from the Language part and "FC" for those from the "Further comments" part. The lines mentioned-below refer to the marked version:

"Language":

All the language issues have been corrected as followed:

1. *"#5: (L. 310 in the track-change document): Replace compile by include."*
   Reply: the word "compile" has been replaced by "include" as requested at the line 298;

2. *"#6: Consistently use mmd□1 instead of mm only. On L. 372, skip from boxplots: the median is not derived from these plots."*
   Reply: as requested, the "mm" have been replaced by "mm/d" from lines 354 to 361, and the expression "median values from boxplots" has been replaced by "median values" only;

3. *"#14: Rephrase as: … period increases $S_{inter}$ values."*
   Reply: as requested, the expression "conducts to increase" has been replaced by "increases" at the line 563;

Further comments:

1. *"#10: Please mention this general trend in the text."*
   Reply: the following sentence has been integrated at the line 473 after "… slightly underestimated.":

   "This behavior is generalizable, being observed on most database sites. However, the phenomenon varies from one site to another without any clear relationship with any soil characteristic, mostly depending on the calibration quality."

2. *"#12: Please specify your statement in the paper according to your response."*
   Reply: the following paragraph has been integrated at the line 437:

   "In this context, a physically-based model is theoretically better suited. However, this kind of model is generally composed of many parameters representing the complexity of a study site such as current crop, root depth, saturated water content (i.e. unsaturated one), hydraulic parameters or water holding capacity. Performing such a model on a large database requires providing all these characteristics for each site, which is in practice very difficult, because the measurement technics are globally expensive. The calibration might be useful but it is time consuming due to the large number of parameters. Conversely, a simple model like SIDRA-RU offers significant advantages, as it requires few information and faster calibration, thus becoming in practice suitable for generalizing."

   The sentence from "As such,…" to "its simple design" has been erased.

3. *"#16: I cannot follow this argument: for each site, the is a value for each parameter from the reference data base and the respective calibrated value. These values can be compared. Please do so. You may add the results in the Appendix."*
Reply: the editor is right, the response for this comment on the previous version was not clear enough. The site-specific comparison cannot be achieved because we do not have the measured hydraulic conductivity K or drainage porosity μ from the calibrated sites. The latter are not included in the list of reference drainage sites. Most of calibrated sites have been settled more recently and were not part of the analysis campaign on the reference drainage sites. The only way to assess the consistency of the calibrated K and μ was to compare the distribution of the calibrated parameters to the distribution of parameters from reference drainage sites, as performed on Fig. 11.

4. *"#19: Please provide the argument also in the paper."*
Reply: the following sentence has been integrated at the line 562 after "a dry calibration period":

"…, defined by a larger occurrence of dry events on a same period due to climate conditions,… ".

We hope this version fulfils your requests.

Best regards,

Alexis Jeantet